# High-throughput screening of caterpillars as a platform to study host–microbe interactions and enteric immunity

Anton G. Windfelder [1,2,3], Frank H. H. Müller [4], Benedict Mc Larney[5,6], Michael Hentschel [7], Anna Christina Böhringer[8], Christoph-Rüdiger von Bredow [9], Florian H. Leinberger[1], Marian Kampschulte[3], Lorenz Maier[7], Yvette M. von Bredow[1], Vera Flocke[10], Hans Merzendorfer[8], Gabriele A. Krombach[11], Andreas Vilcinskas[2,12], Jan Grimm [5,6,13,14,15], Tina E. Trenczek[1,16] ✉ & Ulrich Flögel [10,16] ✉

Mammalian models of human disease are expensive and subject to ethical restrictions. Here, we present an independent platform for high-throughput screening, using larvae of the tobacco hornworm *Manduca sexta*, combining diagnostic imaging modalities for a comprehensive characterization of aberrant phenotypes. For validation, we use bacterial/chemical-induced gut inflammation to generate a colitis-like phenotype and identify significant alterations in morphology, tissue properties, and intermediary metabolism, which aggravate with disease progression and can be rescued by antimicrobial treatment. In independent experiments, activation of the highly conserved NADPH oxidase DUOX, a key mediator of gut inflammation, leads to similar, dose-dependent alterations, which can be attenuated by pharmacological interventions. Furthermore, the developed platform could differentiate pathogens from mutualistic gastrointestinal bacteria broadening the scope of applications also to microbiomics and host-pathogen interactions. Overall, larvae-based screening can complement mammals in preclinical studies to explore innate immunity and host-pathogen interactions, thus representing a substantial contribution to improve mammalian welfare.

Small mammals are the most widely used models to investigate human diseases or to evaluate new imaging technologies. However, mammals have a long generation time, require expensive housing facilities, and there are growing ethical concerns, leading to incorporation of the 3R principle (replacement, reduction, and refinement) into legislation covering animal experiments. The disadvantages of mammalian and other vertebrate models (such as zebrafish or *Xenopus*) can be overcome by the complementary use of insects, which breed rapidly and can be housed in larger numbers at lower costs with less ethical restrictions. Importantly, insects and humans are similar enough for insects to be useful as disease models. More than 60% of the genes in

*Drosophila melanogaster* have human orthologs, many of which are relevant in human diseases[1–4]. Notably, core components of the innate immune system are highly conserved between insects and mammals[5–8]. This includes the insect imd and Toll pathway, which resemble the mammalian TNF-α and TLR pathways[9,10]. There are also similarities in the insulin and (m)TOR pathways[9], as well as in the production of reactive oxygen species (ROS) by the NADPH oxidase dual oxidase (DUOX) during gut inflammation[11]. Eicosanoids promote inflammation in insects and mammals, and both respond to dexamethasone and Cox inhibitors[12,13]. Finally, effector cells of the innate immune system in insects and mammals are comparable, showing

---

similar immuno-metabolic regulations during an immune response[8,14,15], and in both cases, cytokines coordinate the defense against intruders (like bacteria, fungi, and viruses)[5,14]. In particular, *D. melanogaster* has improved our understanding of mammalian innate immunity[16,17]. However, despite its merits, the fruit fly is too small to visualize inflammation, infection, and host–pathogen interactions with imaging modalities such as computed tomography (CT), magnetic resonance imaging (MRI), and positron emission tomography (PET).

To overcome the limitations of small insects, we investigated the use of lepidopteran larvae as a model for human gut inflammation affecting millions of people worldwide[18,19]. Importantly, the midgut epithelial structure and innate immune system of these larvae are histologically and functionally comparable to the human gut[9,20–22]. Here, we employed larvae of the tobacco hornworm *Manduca sexta*, which are big enough for macroscopic imaging techniques, are easy to handle and rear in large numbers. Furthermore, the availability of genetic tools in *M. sexta* helps to elucidate pathological processes down to a molecular level[23,24]. Their compact shape allows the simultaneous imaging of multiple animals in clinical scanners facilitating a true high-throughput screening (Fig. 1). To this end, we used commonly available standard clinical scanners and small animal equipment only for validation studies. We also tested the compatibility of our model with the emerging imaging modality of multispectral optoacoustic tomography (OAT)[25].

In a first step, we established high-throughput imaging of *M. sexta* larvae by CT, MRI, and PET, and then confirmed the detectability of a colitis-like phenotype after bacterial or chemical treatment and its pharmacological rescue. The relevance of this screening platform for preclinical hypothesis testing was demonstrated by exploring the role of the NADPH dual oxidase (DUOX), which is highly conserved between humans and *M. sexta* and aberrantly regulated during gut inflammation in both species[26,27]. In this context, we revealed that DUOX activation led to significant changes in the gut microbiome and showed that our image-based platform could successfully differentiate intestinal bacterial mutualists from pathogens in the gut of *M. sexta*.

## Results

### Verification of intestinal multimodal imaging in *M. sexta* larvae

To assess the gastrointestinal characteristics of *M. sexta*, we followed established imaging protocols for humans and small mammals. In CT and MRI, the primary readouts for tissue integrity are the thickness and signal intensity of the gut wall following extracellular contrast agent (CA) application (Fig. S1a; Supplementary Video 1). In contrast, FDG-PET uses the uptake of [18]F-deoxyglucose (FDG) to evaluate tissue metabolism (Fig. S1b; Supplementary Video 2).

To confirm the visibility of the *M. sexta* gut in clinical CT, MRI, and OAT (Fig. 1), we fed the larvae the appropriate CA, resulting in the clear contrast-enhanced demarcation of the digestive system in coronal (Fig. 1a, b, h, j) and axial views (Fig. 1c, d, i and S2). When CAs for CT and MRI were injected directly into the dorsal vessel (heart equivalent) of *M. sexta* larvae (Supplementary Video 3), we observed a negative contrast in the digestive system, which unequivocally revealed the gut wall. Subsequent experiments confirmed stable contrast enhancement ~15 min after CA application, and we decided to start CT and MRI scans following this interval (Fig. 1m). On axial CT, MRI, and OAT images, the gut wall appeared as a ring-like structure (Fig. 1c, d, i). To verify the selective enhancement of the gut wall, we applied CAs directly to isolated midguts, revealing the same ring-like structure (Fig. 1c, d). We also applied CAs orally and intracardially at the same time (Fig. 1a–d), which enhanced the gut and integument but no other gut-associated organs, such as the fat body (Fig. 2e–j). Finally, first-pass μMRI of the intracardially injected CA (Supplementary Video 4) revealed the distribution of the injected CA over time, which quickly resulted in the clear delineation of the gut wall (Fig. 2b).

Ex vivo μCT demonstrated that the single-layered gut epithelium of *M. sexta* is piled up to form a robust gut wall (Figs. 2a, c and S3; Supplementary Video 5). Furthermore, μCT scans of the midgut demonstrated the validity of the gut wall measurements obtained with the clinical scanner (mean thickness of 1.035 vs. 1.012 mm; Fig. 2d). Next, we used phantoms to confirm that geometric alterations expected during inflammation can be accurately detected by clinical CT and MRI (Figs. S4–S6). Furthermore, we validated the thickness measurements of the gut wall with semi-automatic FWHM thickness measurements in CT and MRI (Figs. S7, S8) and verified the test–retest reliability (S9). Notably, in MRI experiments, the anatomy could be obscured by Gibbs artifacts (Fig. S10), which could be misinterpreted as endoperitrophic space (Fig. S11). Using gut-like phantoms, we found that this artifact was primarily restricted to T2-weighted images and could be minimized by adapting the acquisition matrix (Fig. S12). In optoacoustic imaging, melanin stripes in the integument as well as the thoracic legs and prolegs exhibited high baseline absorptions resulting in large signal intensities. Here, the midgut and heart became apparent following the oral application of optoacoustically active India ink as CA (Fig. 1i–j and S2; Supplementary Video 6).

Additionally, we monitored glucose uptake by *M. sexta* after feeding or injecting FDG followed by PET (Fig. 1e, n). In this case, the gut wall could not clearly be distinguished from the fat body (the larvaes primary metabolic organ) (Fig. 2e–j) when imaging was carried out early after FDG administration, but a gut-shaped pattern began to resolve 1 h after the oral delivery of FDG (Fig. 1n). However, it was not possible to ensure an equal oral uptake of FDG by each animal, so we applied FDG by injection into the dorsal vessel. This required longer exposure times (Fig. 1n), since a gut-shaped pattern became visible only after 3 h. For data evaluation, regions of interest (ROIs) were selected caudal of the head (Fig. 1o) to avoid capturing the strong FDG accumulation in the brain and at injection sites (Fig. 1n, o). For further verification, we dissected individual organs, sampled the hemolymph, and compared the tissue-specific normalized maximal FDG uptake ($PUV_{maxn}$) to the overall $PUV_{maxn}$ in the selected ROI. The results from these experiments confirmed that the head and midgut, but also the fat body, exhibited a high glucose uptake while FDG in the hemolymph was quickly absorbed (Fig. 1p, q).

### Induction of a colitis-like phenotype in *M. sexta*

Next, we characterized the response of *M. sexta* to three different stimuli added to the diet: the pathogen *Bacillus thuringiensis* (Bt) as a positive control for the induction of severe gut inflammation and infection in insects; the chemical dextran sulfate sodium (DSS), which is used for similar experiments in mammals; and a nonpathogenic strain of *Escherichia coli*, which should induce only mild or negligible inflammation. We fed larvae on a normal diet as a negative control. The infection of large fifth-instar larvae (developmental stage L5, day 5/6) with Bt caused a massive increase in lethality compared to the negative control (Fig. S13a, b). Similar results were obtained when smaller second-instar larvae (L2, day 0) were infected with Bt or fed with DSS (Fig. S13c), although the latter caused much milder effects. Larvae fed with *E. coli* showed no difference in survival compared to negative controls (Fig. S13d).

A more detailed characterization of the severe Bt phenotype revealed a significant increase in the mean medial epithelial thickness and gap size (Fig. 3a–f), indicating swelling and fragmentation of the gut epithelium. More plasmatocytes (phagocytes of *M. sexta*) were associated with the medial midgut epithelium in the Bt-infected larvae than in untreated controls (Fig. 3g, h). This was confirmed by a difference in hemocyte numbers between Bt-infected larvae and controls, showing more floating plasmatocytes in the hemolymph (blood) of infected larvae, indicating the rapid onset of an immune response (Fig. 3i, j). Electron microscopy revealed the loss of microvilli, cell swelling, and enterocyte necrosis in the infected larvae but not in the

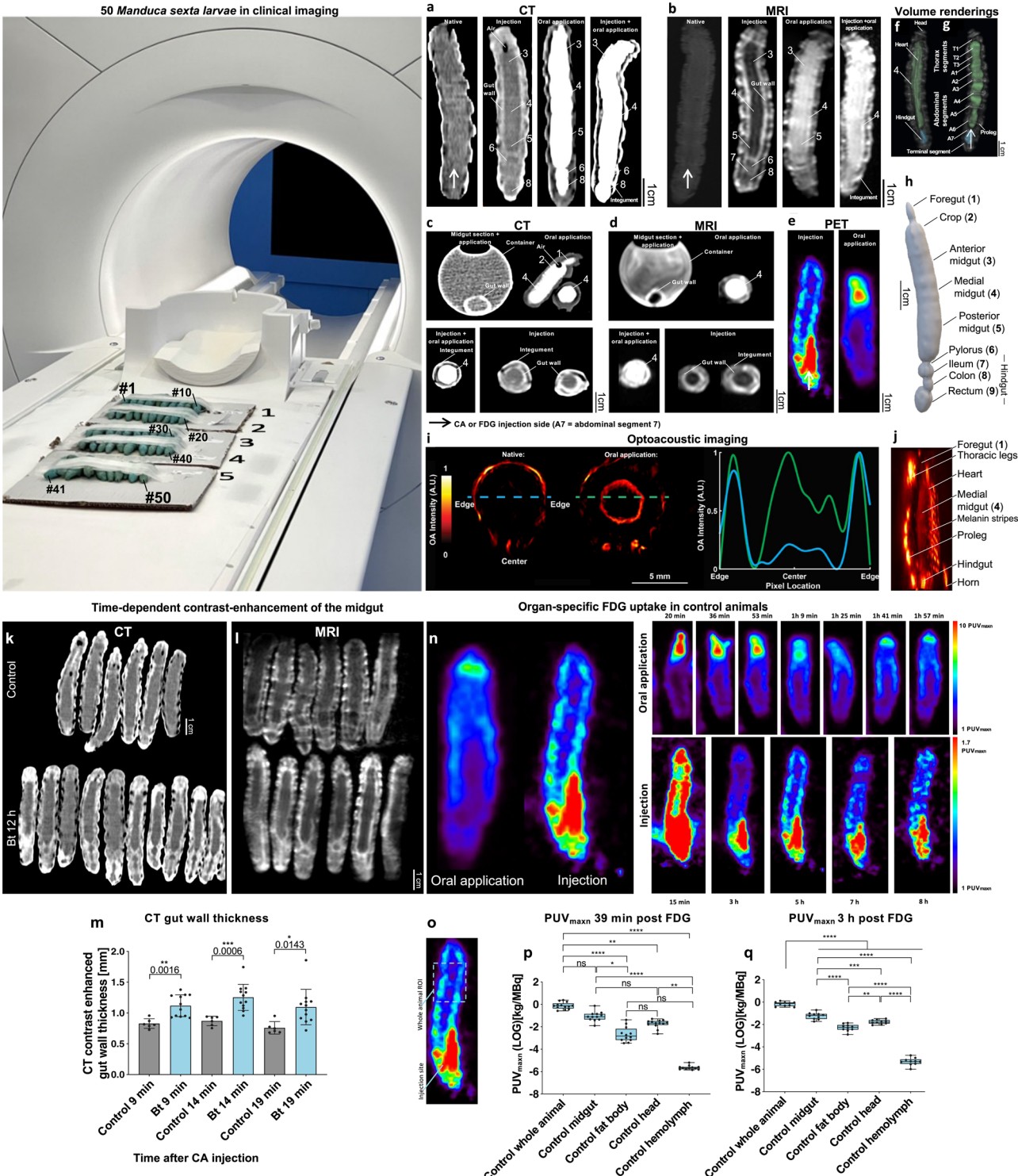

Fig. legend labels within image: 50 *Manduca sexta* larvae in clinical imaging; CT; MRI; Volume renderings; Optoacoustic imaging; Time-dependent contrast-enhancement of the midgut; Organ-specific FDG uptake in control animals; CT gut wall thickness; PUVmaxn 39 min post FDG; PUVmaxn 3 h post FDG.

controls (Figs. 3c, d and S14). We also observed the induction of AMP genes encoding gloverin and attacin-1 in the midgut 24 h after Bt infection (Fig. S15f). Twelve hours later, the bacteria had breached the gut barrier and were detected in the hemolymph (Fig. S15a–c). All Bt-infected larvae showed lesions in the gut wall with strong melanization, a common sign of an immune response in insects (Fig. S15d, e).

Taken together, these data confirmed a severe colitis-like phenotype with substantial structural alterations induced by inflammation and infection, including swelling and the loss of membrane integrity, which should be suitable for detection using CA-based imaging modalities (Fig. S1a).

## High-throughput phenotyping of colitis in *M. sexta* larvae by CT, MRI, and PET

Fifth-instar larvae were selected for screening. After fasting for 1 h, the larvae were allowed to ingest a single feed cuboid loaded with Bt, DSS, or *E. coli*, or normal food as a control. The larvae were then fasted for 12 h to avoid differences in metabolism and were anesthetized on ice for 30 min before CA injection and imaging.

As observed for mammalian gut inflammation, contrast-enhanced CT and MRI scans revealed a thicker gut wall in larvae infected with Bt or fed with DSS than in controls and larvae fed with *E. coli* (Fig. 4a1, b1). A sequential CT analysis in each group demonstrated treatment-

**Fig. 1 | Experimental setup and validation of multimodal imaging of *Manduca sexta* larvae.** 50 *Manduca sexta* larvae were imaged in a clinical MR scanner. The midgut of the *M. sexta* larvae was imaged by CT (**a**, **c**, **h**), MRI (**b**, **d**, **f**, **g**), PET (**e**), and OAT (**i** and **j**: Experiments were repeated independently: $n = 4$ with similar results) with or without the injection and/or oral application of a contrast agent (CA). Axial image of the isolated midgut of *M. sexta* exposed to CA washed out with 0.9% NaCl (**c** and **d**, midgut section + application). Volume rendering of the alimentary tract of *M. sexta* obtained from MRI (**f**, **g**) after injection of Gd-BOPTA, or CT (**h** Experiments were repeated independently: $n = 30$ with similar results) after oral application of iodixanol. Time-dependent contrast-enhancement of the midgut after injection of CA in animals 12 h after Bt infection. **k**, **l** Coronal CT and MR images of control and Bt-infected animals after the injection of CA. **m** CT contrast-enhanced gut wall thickness of control and Bt-infected animals 12 h after Bt infection at 9 min, 14 min, and 19 min after CA injection. The most significant difference between control and Bt infected animals was 14 min after CA injection, $n = 18$, two-tailed t-test. **n** FDG-PET images after oral application or intracardial injections and time-dependent uptake of FDG. **o** Area of $\text{PUV}_{\text{maxn}}$ determination. Organ-specific FDG uptake in control animals. $\text{PUV}_{\text{maxn}}$ values of whole larvae and isolated organs after (**p**) 39 min and ($n = 63$, Kruskal-Wallis-test = 55.1 4, $P = < 0.0001$, box plots: 25th–75th percentiles, whiskers: min-max (show all points), center: median) (**q**) 3 h ($n = 50$, one-way ANOVA, $F(4,45) = 501.0$, $R^2 = 0.978$, $P < 0.0001$, box plots: 25th–75th percentiles, whiskers: min–max (show all points), center: median) of FDG application are shown. The following significance levels have been used: ns $= P > 0.05$, * $= P \le 0.05$, ** $P \le 0.01$, *** $= P \le 0.001$ and **** $= P \le 0.0001$. Bar charts represent mean and SD. Every data point represents a single animal. Source data are provided as a Source Data file.

dependent thickness curves all along the gut ($F = 391$ (3,301), $p < 0.0001$; Fig. 4d). Interestingly, only animals infected with Bt showed a high CT signal density or normalized T1-weighted signal in the gut wall compared to all other groups (Fig. 4a2, b2). In Bt-infected animals, the gut wall signal intensity was also significantly higher in native T2-weighted MRI scans without CA indicating inflammation-associated edema (Fig. 4b3, b4). Again, this effect was restricted to the Bt group and was not observed in DSS-treated animals.

All CT and MRI findings described above are consistent with mammalian inflammatory symptoms, but the assessment of colitis-induced larval metabolism by PET yielded surprising results. Typically, inflamed tissue takes up more glucose due to the high glycolytic turnover of infiltrated immune cells. However, we observed the opposite: FDG-PET revealed a significantly lower $\text{PUV}_{\text{maxn}}$ in the apical region of animals fed with Bt or DSS compared to the control and *E. coli* groups (Fig. 4c1). Because the midgut and fat body could not be distinguished, we dissected these organs from Bt-infected and control animals and checked them for tissue-specific inflammation-dependent differential FDG uptake, confirming that the $\text{PUV}_{\text{maxn}}$ of these organs in Bt-infected animals was lower than the control value (Fig. S16). The severe Bt phenotype with incipient necrosis (Figs. 3d and S14) probably leads to an overall depression of FDG uptake[28] and masks the inflammation-associated effects.

Although we always used animals at the same stage of development, we wanted to exclude the possibility that size differences affected the imaging data. We, therefore, calculated general linear models for each imaging modality (CT $n = 75$, MRI $n = 92$, and PET $n = 88$) which confirmed that animal treatment but not size had a significant effect on CT and MRI contrast-enhanced gut wall thickness or $\text{PUV}_{\text{maxn}}$ values (Table S2). The sensitivity and specificity of all findings were calculated and compared to corresponding data for human Crohn's disease (Tab. S3, last column). The CT contrast-enhanced gut wall thickness and normalized MRI T1 signal showed the highest sensitivities and specificities, with ROC curve areas of 0.97 and 0.91, respectively, followed by MRI contrast-enhanced gut wall thickness (0.86), CT signal density (0.8), MRI T2 signal (0.78), and $\text{PUV}_{\text{maxn}}$ (0.72). Additionally, we analyzed the effect size of every treatment (Tab. S14–S17) and identified T1 signal enhancement as parameter with the largest value ($g = 6.7$, Tab. S14), rendering it as an excellent readout for high-throughput screenings. Furthermore, multiple chi-squared tests were applied to the data (Fig. S17), revealing that Bt-infected animals exceeded the defined thresholds for all six imaging findings, DSS-fed animals in four cases, and animals exposed to *E. coli* only in one case. Finally, we found significant correlations between the different diagnostic features, providing additional verification of their results (Fig. 5a–e).

## Temporal progression and antimicrobial treatment of colitis in *M. sexta*

In a sub-cohort of animals, we monitored the impact of Bt infection beyond the 12 h observation period applied above. After 36 h, almost all CT and MRI measures revealed a severe aggravation of the larvae phenotype compared to the 12 h post-infection findings (Fig. 5f–l). The most prominent alterations were observed for the MRI T1 signal, which approximately doubled compared to the signal at 12 h, indicating progressive structural damage. At the metabolic level, FDG uptake decreased further over time, confirming progressive tissue impairment due to necrosis (Fig. 5f).

Next, we evaluated our model for screening of antimicrobial drugs or new pharmaceutical concepts. Accordingly, we applied gentamicin to the larvae infected with Bt because this antibiotic is known to attenuate the toxicity of Bt in lepidopterans[29]. We then monitored the effects of the treatment on gut wall thickness by contrast-enhanced CT and MRI, because these imaging methods showed high sensitivity and specificity for the assessment of Bt and other treatments in *M. sexta* (Tab. S3; Fig. 4). Both CT and MRI revealed that gentamicin treatment had a concentration-dependent effect on gut wall thickness, resulting in values comparable to those of control larvae at the highest doses (Fig. 4e, f). Importantly, this effect was also reflected in the differential survival of larvae fed with Bt only, Bt with low/high gentamicin treatment, and control animals (Fig. 4g).

## Characterization of DUOX and the impact of its activation in *M. sexta*

DUOX is a transmembrane protein in the gastrointestinal tract of humans and most other animals (Fig. S18c). Beyond its physiological importance in tyrosine cross-linkage, DUOX plays an essential role in the mucosal immunity of the gut[30,31]. It is activated by bacterially-derived uracil[32–34] and forms HOCl as well as other ROS that attack pathogens (Fig. 6a)[33,34]. Homology modeling predicted that the tertiary structure of DUOX and especially its peroxidase homology domain (PHD) is highly conserved between *M. sexta* and humans (Figs. 6b–c, and S19; Supplementary Videos 7–8). Importantly, a heme-binding site within the PHD, which is required for peroxidase activity, was predicted in human DUOX1 and DUOX2 as well as the *M. sexta* and *D. melanogaster* enzymes (Fig. S20b; Tables S5–S8; Supplementary Video 9). In line with these predictions, we found that exposure of *M. sexta* to uracil led to enhanced HOCl production and an upregulation of gastrointestinal DUOX (Fig. 6d and Figs. 7b, c, d, and S22, S23). Also similar to humans, immunohistochemical analysis of *M. sexta* sections revealed the localization of DUOX in the apical region of enterocytes (Figs. 7a and S21)[35]. Following oral uracil treatment, we observed a rapid decline in colony-forming units (Fig. 7i) and a shift in bacterial community composition towards the more HOCl-resistant *Enterococcus* sp. #4 in the fecal microbiota (Fig. 9a, b and h, i), highlighting the potent antibacterial properties of HOCl. However, this also affected *M. sexta* itself, as shown by the concentration-dependent effect of uracil on survival and weight gain (Fig. 7e, f and g, h). This was accompanied by intestinal cell swelling, disruption/loss of microvilli, and general damage to the intestinal epithelial surface as revealed by electron microscopy (Fig. S24). These ultrastructural findings were associated with macroscopic melanized lesions that ranged from small

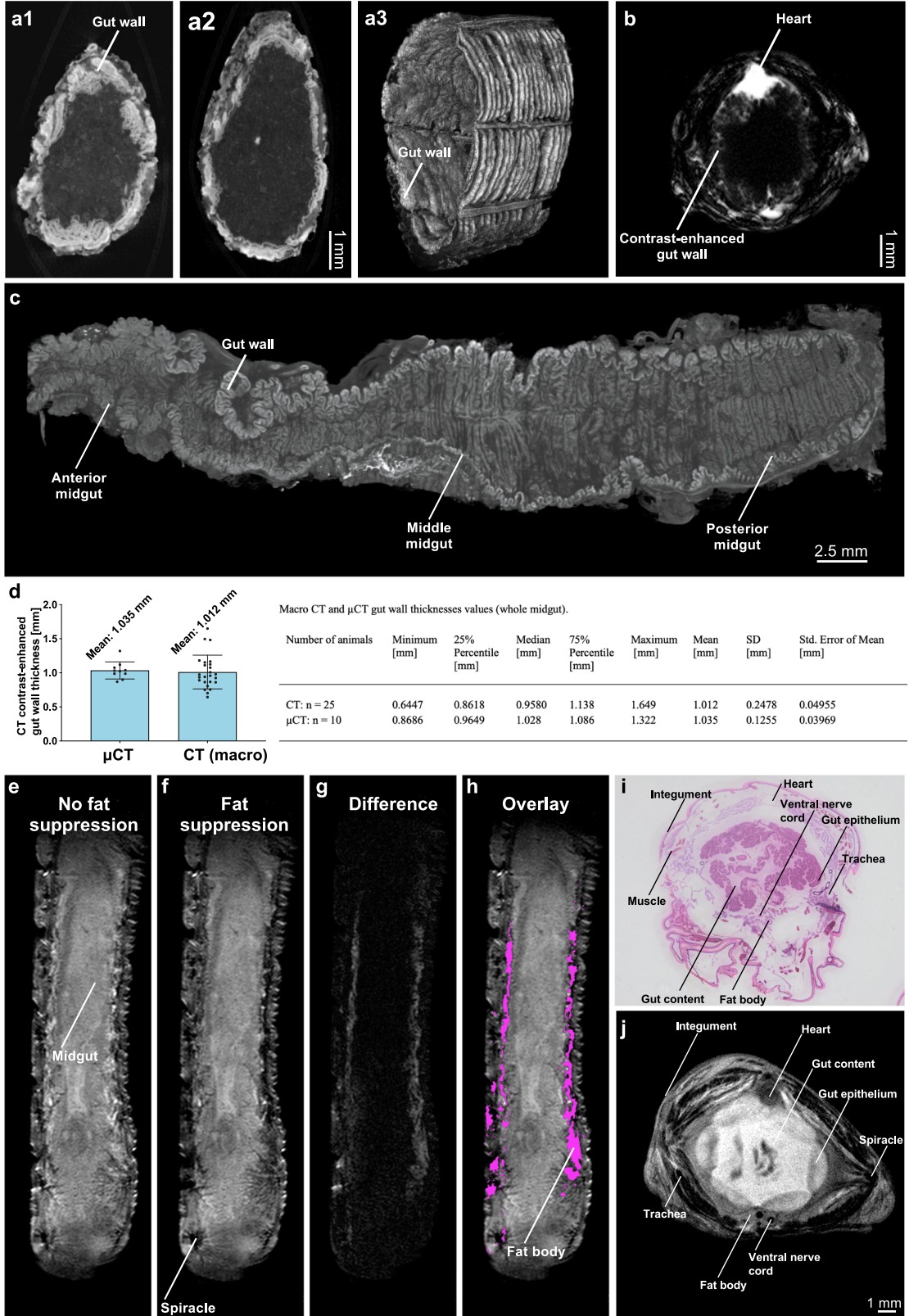

**d** Macro CT and µCT gut wall thicknesses values (whole midgut).

| Number of animals | Minimum [mm] | 25% Percentile [mm] | Median [mm] | 75% Percentile [mm] | Maximum [mm] | Mean [mm] | SD [mm] | Std. Error of Mean [mm] |
|---|---|---|---|---|---|---|---|---|
| CT: n = 25 | 0.6447 | 0.8618 | 0.9580 | 1.138 | 1.649 | 1.012 | 0.2478 | 0.04955 |
| µCT: n = 10 | 0.8686 | 0.9649 | 1.028 | 1.086 | 1.322 | 1.035 | 0.1255 | 0.03969 |

isolated areas in the anterior midgut to extensive lesions all over the digestive tract depending on the uracil dose (Fig. S25).

To explore the relationship between DUOX and ROS in more detail, we supplemented the food cuboids with diphenyleneiodonium (DPI) as bona fide NOX inhibitor[36] and N-acetyl-ʟ-cysteine (NAC) as ROS scavenger[37]. DPI plus uracil in the food reduced HOCl production to baseline levels (Fig. 6d), confirming the DUOX-dependent generation of HOCl in response to uracil. Although NAC did not reduce uracil-induced HOCl production (Fig. 6d), the survival rates and growth of animals treated with uracil and NAC improved (Fig. 7f–h). The latter was also true for DPI, even though the long-term exposure of *M. sexta* to this inhibitor worsened its survival (Fig. 7f). Putative DPI

**Fig. 2 | High-resolution verification of in vivo measurements of multimodal imaging of *M. sexta* larvae. a1–a3** Contrast-enhanced ex vivo μCT of two different anterior midgut regions (smaller diameter and larger diameter) and with a 3D volume rendering (Supplementary Video 5). **b** Axial in vivo μMRI of a *M. sexta* L5 larva 10 min after the injection of contrast agent (CA). The gut wall shows the strong uptake of CA. **c** PTA-stained midgut in μCT, **d** Mean CT (*n* = 25, voxel size of 350 μm) and μCT (*n* = 10, voxel size of 7.55 μm) gut wall thicknesses from the complete midgut with descriptive statistics. The fat body of *M. sexta* in μMRI with (**f**) and without (**e**) fat suppression. The difference between e and f is shown in **g** and corresponds to the fat body of *M. sexta* (**h**). **i** Shows an axial paraffin section (L5d2), which shows the fat body and confirms our analysis of **e** and **f**. **j** This shows a corresponding axial μMRI of an *M. sexta* (L5d5) larvae. Bar charts represent mean and SD. Every data point represents a single animal. Source data are provided as a Source Data file.

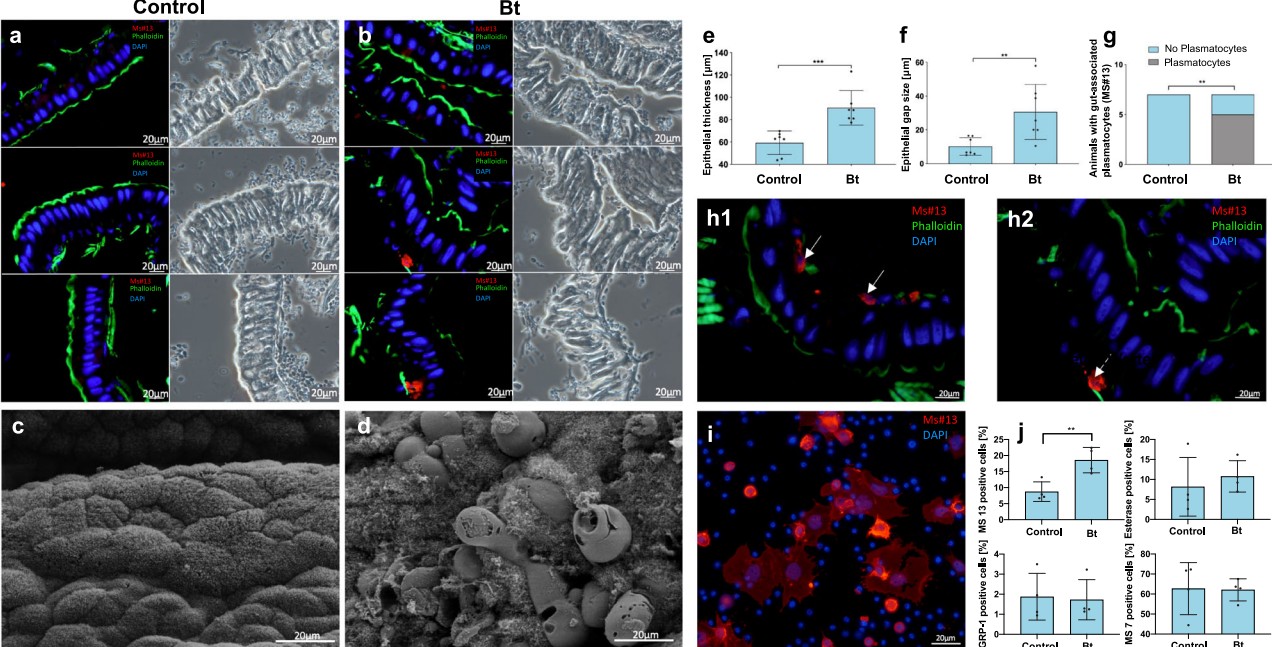

**Fig. 3 | Histopathological characterization of the colitis-like phenotype in larvae treated with *Bacillus thuringiensis* (Bt). a**, **b** Midgut histology (cryosections) from control (*n* = 7) and Bt-treated (*n* = 7) larvae. On the left are fluorescence microscopy (FM) images, on the right the corresponding phase-contrast microscopy (PCM) images. Bt-infected animals differed significantly in gut wall thickness (two-tailed Mann Whitney test, *p* = 0.0006, no adjustments, **e** and gap size (two-tailed Mann Whitney test, *p* = 0.0041, no adjustments **f** and had gut-associated plasmatocytes (red, monoclonal antibody Ms#13) (two-sided Chi-square-test, *P* = 0.0053, **g**). **h1** and **h2** are showing the gut-associated plasmatocytes from Bt-treated animals, indicated by arrows. F-actin is stained with FITC-phalloidin (green), indicating microvilli on the apical side of the epithelium. Nuclei are counterstained with DAPI (blue). **c**, **d** Midgut ultrastructure (scanning electron microscopy) from control (*n* = 5) and Bt-treated (*n* = 3) larvae. Bt-exposed larvae showed loss of microvilli, cell swelling, and necrosis. **i** Changes in differential hemocyte count 12 h after Bt infection. Plasmatocytes were labeled with antibody MS#13 (red) and the nuclei of all hemocytes are counterstained with DAPI (blue). **j** Animals infected with Bt (*n* = 4) featured a significantly higher number of floating plasmatocytes than control animals (*n* = 4), indicating a change in the cellular immune response (two-tailed t-test, *p* = 0.0076, no adjustments). No changes were observed in spherule cells (esterase positive), oenocytoids (GRP-1 positive), or granular cells (MS#7 positive). The following significance levels have been used: ns = *P* > 0.05, * = *P* ≤ 0.05, ** = *P* ≤ 0.01, *** = *P* ≤ 0.001 and **** = *P* ≤ 0.0001. Bar charts represent mean and SD. Every data point represents a single animal. Source data are provided as a Source Data file.

cytotoxicity[38] was also observed in animals treated with DPI alone, but not those treated with NAC alone (Fig. 7f).

## High-throughput screening of DUOX activation in *M. sexta*

Next, we used high-throughput CT, MRI, and FDG-PET to characterize the DUOX-related phenotype (Fig. 8). The primary imaging readouts established above (gut wall thickness and signal intensity) clearly revealed that DUOX activation by uracil caused effects similar to those previously observed for Bt and DSS (Fig. 8a–f). Interestingly, FDG-PET analysis revealed a higher $PUV_{maxn}$ for the uracil-treated *M. sexta* larvae, and subsequently, a new threshold value was defined (Fig. 8c1, Table S9). The phenotype induced by uracil was milder than that induced by Bt and comparable to the modest effect of DSS. Electron microscopy revealed no evidence of necrosis (Fig. S24), which could mask the increased FDG uptake by infiltrated immune cells[8,14,15,28], suggesting that the high glycolytic turnover of recruited immune cells dominates the $PUV_{maxn}$[8,14,15].

Notably, DPI and NAC almost completely rescued the uracil-induced phenotype (Fig. 8a–d), strongly supporting the DUOX-HOCl

axis as the major underlying pathophysiological mechanism (Fig. 8a–d, Fig. S26). Again, sequential CT analysis demonstrated treatment-dependent thickness curves all along the gut (*F* = 330.2 (6,535) *p* < 0.0001; Fig. 8d). Finally, dexamethasone treatment prevented the uracil-induced colitis-like phenotype in *M. sexta*, further corroborating the screening value of our platform (Fig. 8f–f1).

Gut wall thickness assessed by contrast-enhanced CT was already the most precise imaging readout during Bt infection and also sensitively distinguished between different uracil concentrations (Fig. 8a3). MRI was less responsive to dose-dependent effects (Fig. 8b3), probably due to its lower resolution, whereas high uracil doses blunted the rise in $PUV_{maxn}$ during FDG-PET analysis reported above (Fig. 8e). This is similar to our observations during severe Bt infection (Fig. 4c1) and may again reflect the metabolic depression of non-immune tissue and the onset of necrosis due to enhanced ROS generation.

Overall, these findings demonstrate that our approach is suitable to monitor the appearance, progression, and treatment of colitis-like inflammatory patterns evoked by excessive DUOX activation in

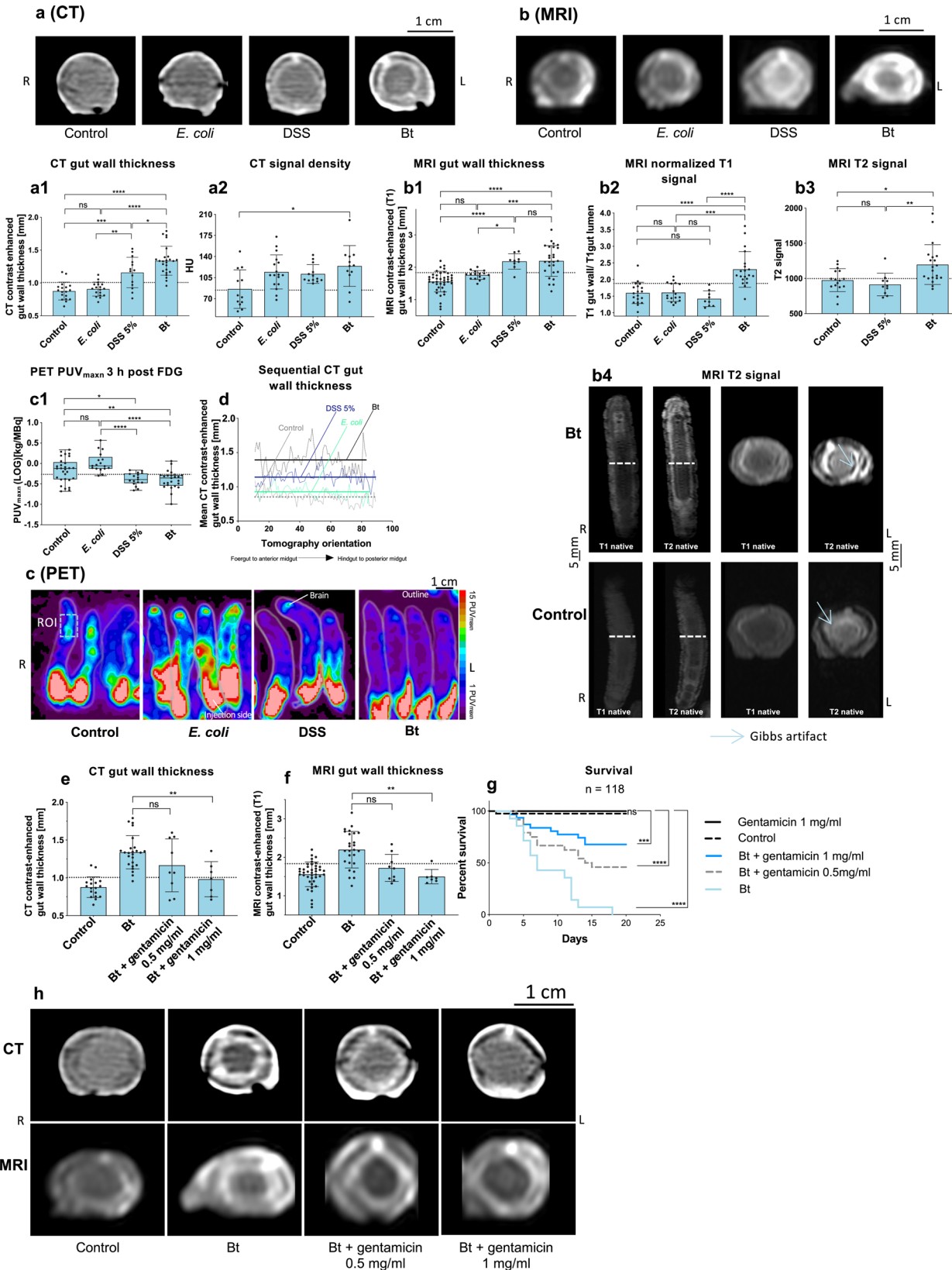

*M. sexta* and is therefore useful for preclinical hypothesis testing and high-throughput screening of new therapeutic concepts.

### Assessing host–pathogen interactions in the gut

In a final step, we extended our approach to the analysis of host-pathogen interactions. To this end, we investigated the

microbiome of *M. sexta* by culture-dependent 16S rRNA sequencing of fecal pellets, identifying two dominant enterococci (*Enterococcus casseliflavus/gallinarum* #1 and *Enterococcus* sp. #4) and a *Microbacterium* species (Fig. 9a, b). These results are also in line with previous studies (Tab. S11) reporting the microbiome of *M. sexta* of low complexity[39–41]. Interestingly, oral

**Fig. 4 | High-throughput imaging of larvae exposed to different challenges and concentration-dependent rescue of the colitis-like phenotype with gentamicin.** *M. sexta* larvae shown in axial CT (**a**), MRI (**b**), and PET (**c**). Animals were fed with a control diet or a diet containing *Escherichia coli*, DSS, or *Bacillus thuringiensis* (Bt). **a1–c1** Quantification of the imaging data, a1: $n = 75$, one-way ANOVA, $F(3,71) = 30.23$, $R^2 = 0.5609$, $P < 0.0001$, a2: $n = 54$, Kruskal-Wallis-test = 9.954, $P = 0.019$, b1: $n = 92$, one-way ANOVA, $F(3,88) = 21.72$, $R^2 = 0.4255$, $P < 0.0001$, b2: $n = 65$, Kruskal-Wallis test = 31.74, $P < 0.0001$, b3: $n = 46$, Kruskal-Wallis test = 13.58, $P = 0.0011$ and c1: $n = 88$, one-way ANOVA, $F(3,84) = 13.15$, $R^2 = 0.3196$, box plots: 25th–75th percentiles, whiskers: min–max (show all points), center: median. **d** Mean sequential CT gut wall thickness curves (the horizontal lines represent the

overall treatment-specific mean thickness, preferred model: different curve for each data set, $F(3, 301) = 391$, $P < 0.0001$. Contrast-enhanced CT and MRI gut wall thickness showed a gentamicin concentration-dependent reduction of gut wall thickness in animals fed with Bt and two different gentamicin concentrations (**e**): $n = 59$, one-way ANOVA, $F(3,55) = 15.83$, $R^2 = 0.4634$, $P < 0.0001$, (**f**): $n = 83$, Kruskal-Wallis-test = 28.07, $P < 0.0001$. This finding was also confirmed by the differential survival of the treated animals (**g**). Dashed lines indicate threshold values. Significance levels: ns = $P > 0.05$, *$P \le 0.05$, **$P \le 0.01$, ***$P \le 0.001$ and ****$P \le 0.0001$. Survival kinetics shows the sum of the conducted experiments. Bar charts represent mean and SD. Every data point represents a single animal. Source data are provided as a Source Data file.

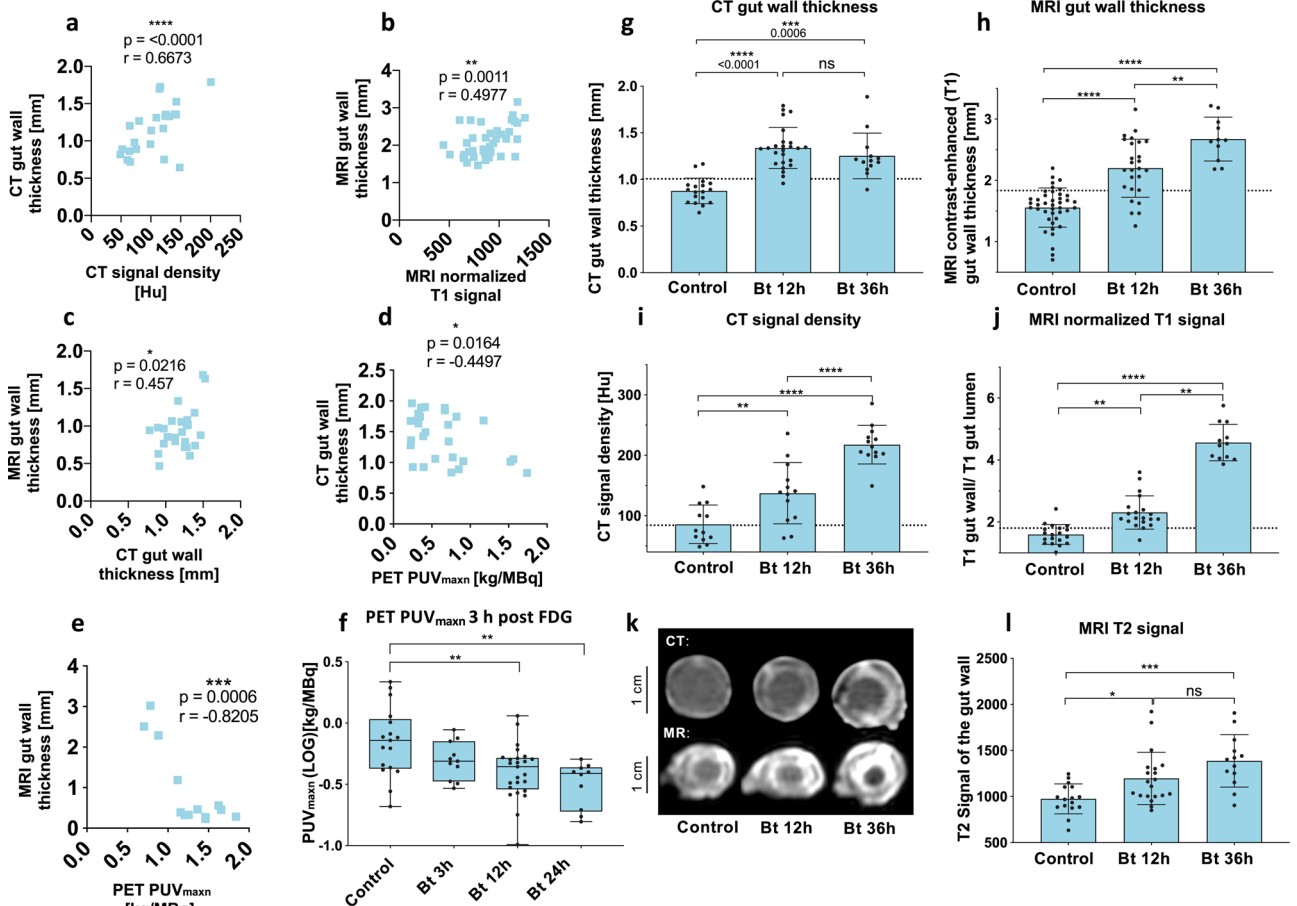

**Fig. 5 | Correlations of diagnostic findings and temporal development of the colitis-like phenotype in *M. sexta*.** Correlations (Pearson) of the used diagnostic findings with each other. (**a**) Relationships between the two CT features (maximum CT-contrast-enhanced gut wall thickness and maximum signal density of the gut wall, $n = 32$, $r = 0.6673$, 95% CI = 0.4153–0.8242, $r^2 = 0.4453$ and two-tailed $P < 0.0001$. **b** Correlations between two MRI features (MRI-contrast-enhanced gut wall thickness and normalized T1 signal, $n = 40$, $r = 0.4977$, 95% CI = 0.2204–0.7006, $r^2 = 0.2477$ and two-tailed $P = 0.0011$). **c** Correlations between the CT feature (CT-contrast-enhanced gut wall thickness) and the MRI feature (MRI-contrast-enhanced gut wall thickness), $n = 25$, $r = 0.457$, 95% CI = 0.07551–0.7218, $r^2 = 0.2089$ and two-tailed $P = 0.0216$. **d** Correlations between the CT feature (CT-contrast-enhanced gut wall thickness) and the PET feature (PUVmaxn), $n = 28$, $r = -0.4497$, 95% CI = −0.7046 to −0.0924, $r^2 = 0.2022$ and two-tailed $P = 0.0164$. **e** Correlations between the PET feature PUV$_{maxn}$ (maximum standard uptake value) and the MRI features (MRI-contrast-

enhanced gut wall thickness), $n = 13$, $r = -0.8205$, 95% CI = −0.9445−0492, $r^2 = 0.6733$ and two-tailed $P = 0.0006$. Temporal development of the colitis-like phenotype in *M. sexta* at baseline conditions and after Bt treatment after 3 h, 12 h, and 24 h or 36 h in **f** PET, $n = 63$, one-way ANOVA, $F(3,59) = 6.064$, $R^2 = 0.2357$, $P = 0.0011$, box plots: 25th–75th percentiles, whiskers: min–max (show all points), center: median, **g** and **i** CT, $n = 56$, Kruskal-Wallis test = 32.54, $P < 0.0001$ and $n = 38$, one-way ANOVA, $F(2,35) = 35.66$, $R^2 = 0.6708$, $P < 0.0001$, and **h**, **j**, and **l** MRI, $n = 78$, one-way ANOVA, $F(2,75) = 46.52$, $R^2 = 0.5537$, $P < 0.0001$, $n = 51$, Kruskal-Wallis test = 38.45, $P < 0.0001$ and $n = 49$, Kruskal-Wallis test = 17.36, $P = 0.0002$. The following significance levels have been used: ns = $P > 0.05$, * = $P \le 0.05$, ** $P \le 0.01$, *** = $P \le 0.001$ and **** = $P \le 0.0001$. The dashed lines in the diagrams illustrate the respective threshold values received from the ROC analysis. Error bars represent standard deviation. Bar charts represent mean and SD. Every data point represents a single animal. Source data are provided as a Source Data file.

inoculation of *M. sexta* with the isolated bacterial species revealed the *Microbacterium* species as potent pathogen leading to similar CT and FDG-PET patterns as Bt infection (Fig. 9c, d) accompanied by drastically reduced survival (Fig. 9e, g). In parallel, we observed that the supernatant of both *Enterococcus*

species not only suppressed the growth of the pathogenic *Microbacterium* species (Fig. 9f, j) but also rescued *M. sexta* fed with the *Microbacterium* species (Fig. 9g), revealing the co-existence of these species as a common example of intestinal immunoprotective symbionts in Lepidoptera[42]. Of note, also an

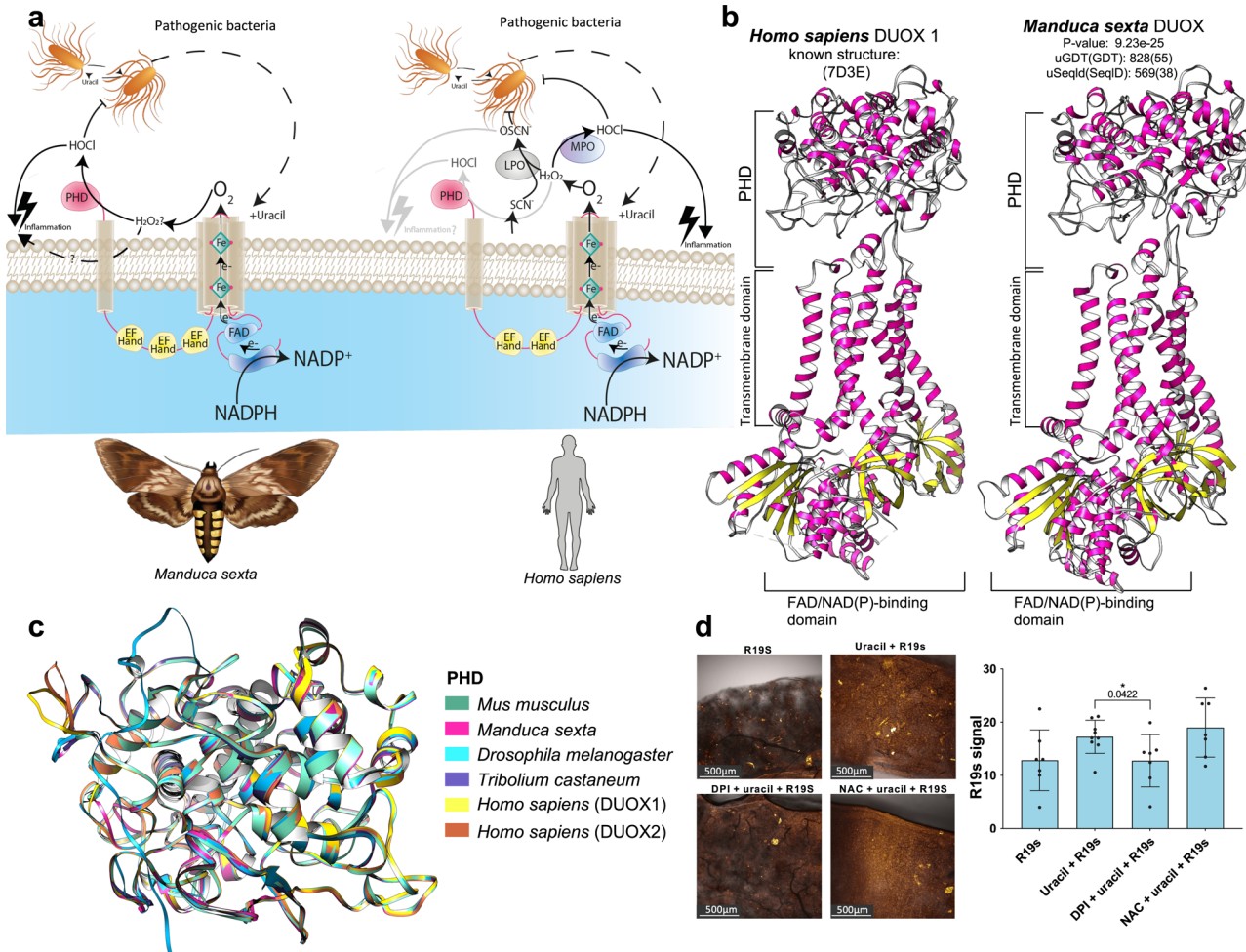

**Fig. 6 | Conservation of human and *M. sexta* DUOX and HOCl production.**
**a** Comparative structures of DUOX in *M. sexta* and *Homo sapiens* (DUOX2).
**b** Homology modeling of *M. sexta* DUOX. The known structure of *H. sapiens* DUOX1 (left) and model-assisted protein binding site prediction of *M. sexta* DUOX (right). Regions that were not modeled in the human DUOX1 template structure are also not shown in the model of *M. sexta* DUOX. **c** The peroxidase homology domain (PHD) is the active site of HOCl production and is predicted to be highly conserved by model-assisted protein binding site prediction. **d** DUOX-dependent HOCl production after uracil treatment and its inhibition with diphenyleneiodonium (DPI) but not with N-acetylcysteine (NAC), $n = 16$, two-tailed t test. Bar charts represent mean and SD. Every data point represents a single animal. Source data are provided as a Source Data file. $n = 30$, one-way ANOVA, $F(3,26) = 3.084$, $R^2 = 0.2624$, $P = 0.0448$. Bar charts represent mean and SD. Every data point represents a single animal. Source data are provided as a Source Data file.

antibiotic treatment of untreated larvae led to a better development as reflected by a significantly increased weight gain than in control animals (Fig. 9k, l).

The metabolic characterization of the isolated bacteria (Table S12) revealed that *Enterococcus casseliflavus/gallinarum* #1 used casein and starch, while *Enterococcus* sp. #4 was able to use cellulose as a primary energy source. Accordingly, the isolated enterococci were able to metabolize the main components of the artificial *M. sexta* diet (Source Data file)[43,44]. Taken together, these results demonstrate a possible trade-off for hosting an immunoprotective mutualist such as *Enterococcus casseliflavus/gallinarum* #1 and *Enterococcus* sp. #4 with sharing of the nutritional resources between the host and its mutualists.

In separate experiments, we further demonstrated the importance of the uracil-DUOX-HOCl-axis for the microbiome in *M. sexta* in that *Enterococcus* sp. #4 was significantly more resistant against HOCl than *Enterococcus casseliflavus/gallinarum* #1 (Fig. 9i, Table S12). As a consequence, although uracil treatment led to a decline of the total bacterial count (Figs. 9b and 7i), the proportion of *Enterococcus* sp. #4 in the overall appearance increased significantly (Fig. 9h). Therefore, our approach is also suitable to identify intestinal pathogens as well as host-commensal interactions in the gut microbiome.

## Discussion

We have demonstrated that routine clinical imaging procedures are suitable for the large-scale phenotyping of *M. sexta* as a toolbox to study distinct aspects of innate immunity in the gut. By exposing larvae to defined challenges, we showed that multimodal imaging can be used to assess the severity and progression of gut inflammation and infection with great sensitivity, allowing the identification of novel therapeutic targets and evaluation of new pharmacological concepts in preclinical screening studies.

We initially achieved proof of concept for our model by imposing bacterial or chemical challenges, followed by CT, MRI, and FDG-PET analysis for the detection of colitis. Statistical analysis revealed similar sensitivities and specificities in the *M. sexta* model as for human Crohn's disease (Table S3)[45,46]. In our model, contrast-enhanced CT was more sensitive than MRI for gut wall thickness measurements, which probably reflects the lower resolution of the clinical MRI scanner. The MRI slice thicknesses we used resulted in an overestimation of gut wall thickness, but the absolute gut wall thickness measured by CT agreed with µCT gut wall thickness measurements (Fig. 2d). Importantly, the differences in gut wall thickness between Bt, DSS, uracil or *Microbacterium* sp. treatments and the control treatment were greater than

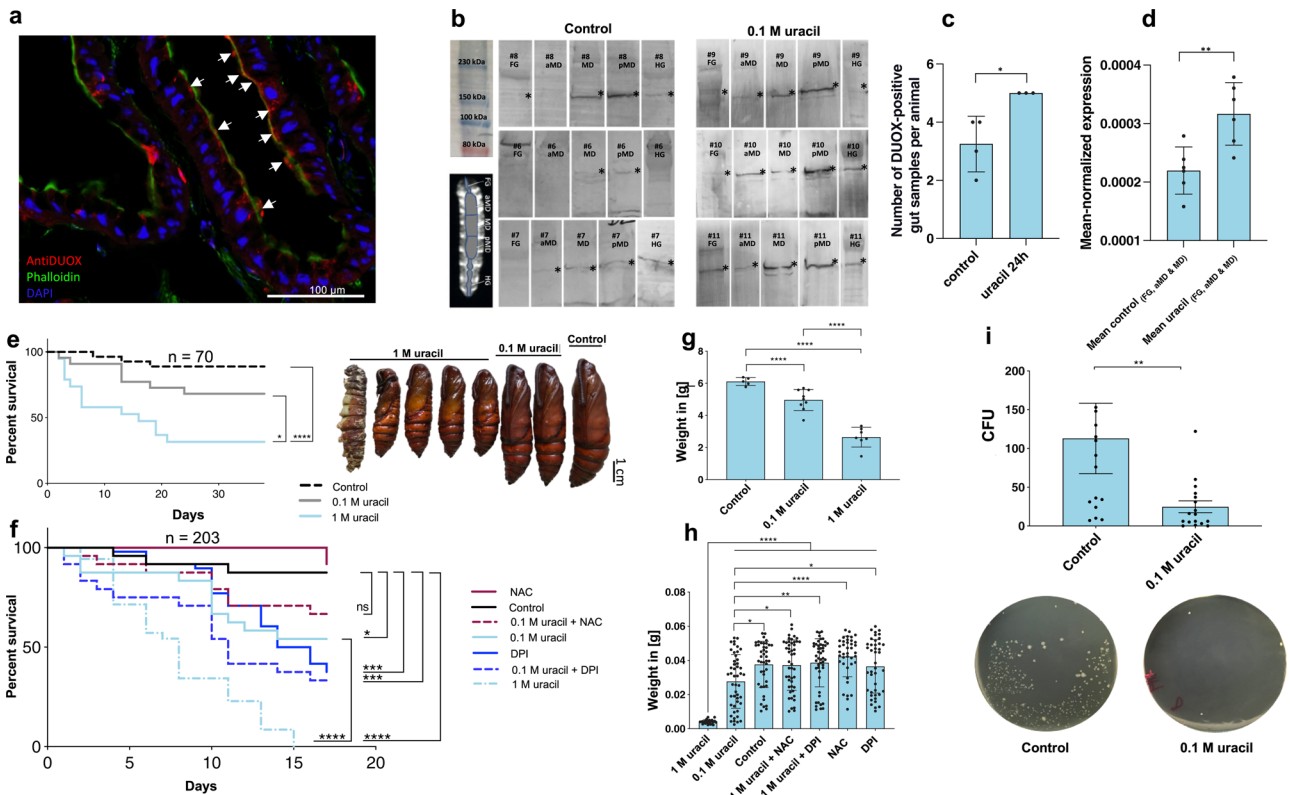

**Fig. 7 | Location of *M. sexta* DUOX in the gut, its upregulation, and effect. a** *M. sexta* DUOX is localized in the brush border of the midgut, experiments were repeated independently: $n = 8$ with similar results. **b**, **c** Western blots confirming the upregulation of *M. sexta* DUOX after uracil treatment (24 h). FG: foregut, aMD: anterior midgut, MD: medial midgut, pMD: posterior midgut, and HG: hindgut. * indicates a protein with the rel. molecular weight of 170 kDa (DUOX). More details about the Western blot analysis are given in Fig. S22, two-tailed Mann Whitney test, $P = 0.0286$, no adjustments, experiments were repeated independently: $n = 7$ with similar results. See the source data file for a presentation of complete, uncropped blots and gels. **d** Quantitative real-time RT-PCR analysis of *MsDUOX* demonstrating the induction of *M. sexta DUOX* after uracil treatment (8 h) in the anterior digestive tract (FG, aMD, and MD). More details about the qPCR analysis are given in Fig. S23, $n = 12$, two-tailed $t$ test, $P = 0.0054$, no adjustments. **e–h** Concentration-dependent influence of DUOX (activated by uracil) on survival and weight, **g** $n = 21$, one-way ANOVA, $F_{(2,18)} = 58.97$, $R^2 = 0.8676$, $P < 0.0001$, **h** $n = 288$, one-way ANOVA, $F_{(6,281)} = 24.91$, $R^2 = 0.3472$, $P < 0.0001$. **i** Bactericidal effect of HOCl after uracil treatment, resulting in fewer CFUs in the feces of animals treated with uracil, two-tailed Mann Whitney test, $P = 0.0037$, no adjustments. Survival kinetics shows the sum of the conducted experiments. Bar charts represent mean and SD. Every data point represents a single animal. The following significance levels have been used: ns $= P > 0.05$, $* = P \le 0.05$, $** = P \le 0.01$, $*** = P \le 0.001$ and $**** = P \le 0.0001$. Source data are provided as a Source Data file.

the doubled mean empirical standard deviation for MRI and CT phantom thickness measurements (Tables S2, S10, S13; Fig. S4–S6). These measurements were validated via semi-automatic FWHM thickness measurements, and their reliability was confirmed by test-retest analysis (Fig. S7–S9). Of note, the disease models used in this study induced moderate to severe alterations in gut wall thickness as compared to control animals which could easily be resolved by our current approach. However, if the induced effects are weaker, optimization of the resolution may be required. For this, the larvae's long and straight body shape allow densely packed experimental setups with small FOVs (Fig. 1). Furthermore, instead of averaging all CT and MRI parameters over the entire midgut (as in the present study), a regional or sub-regional analysis might be advantageous if the colitis-like phenotype is restricted to certain midgut areas. In PET, the resolution problem was more challenging, but here it is advantageous that the entire larval metabolism seems to respond to the induced inflammation during severe challenges[8]. Beyond those conventional imaging modalities, we have also shown that new approaches such as OAT are compatible with our model and can accommodate even smaller larval stages of *M. sexta* (Figs. 1i–j and S2; Supplementary Video 6).

To evaluate the severity and likelihood of gut inflammation, we determined threshold values for each imaging procedure. As expected, there were significant differences in all diagnostic parameters when comparing Bt-infected and control animals (Fig. S17). However,

for DSS-fed animals, these threshold values differed in only four of the six imaging findings and for *E. coli* solely in one (Fig. S17). These results demonstrate the decreasing severity of inflammation from Bt (severe) and DSS (moderate) to *E. coli* (slight to negligible) and mirror the corresponding survival kinetics. The concentration-dependent rescue of animals fed with Bt by gentamicin emphasizes our model's potential to identify new immunomodulatory or antimicrobial compounds.

The fundamental immunologic, metabolic, and pathological processes of inflammation in insects are analogous to those in mammals, facilitating analysis by contrast enhancement with extracellular CAs and aberrant FDG uptake. Interestingly, the infection of *D. melanogaster* with a parasitoid wasp led to a lower global [14]C-labeled glucose uptake in the fat body and other tissues but to a higher uptake in hemocytes[8]. Thus, during severe disease states, the global metabolic suppression of the fat body and incipient necrosis (Figs. 3d and S16) most likely disguise locally increased inflammation-dependent FDG uptake by hemocytes[8,14,15,28]. Furthermore, sugar metabolism is somewhat different in insects and mammals: glucose in insects is temporarily converted to trehalose, which may also contribute to inconsistent PET findings. On the other hand, an important specific aspect of our model is the open circulatory system of insects. Accordingly, the distribution of the injected CA occurs slowly and evenly throughout the larvae (Supplementary Videos 3 and 4) and is not determined by vessel topography or vascular permeability. Hence,

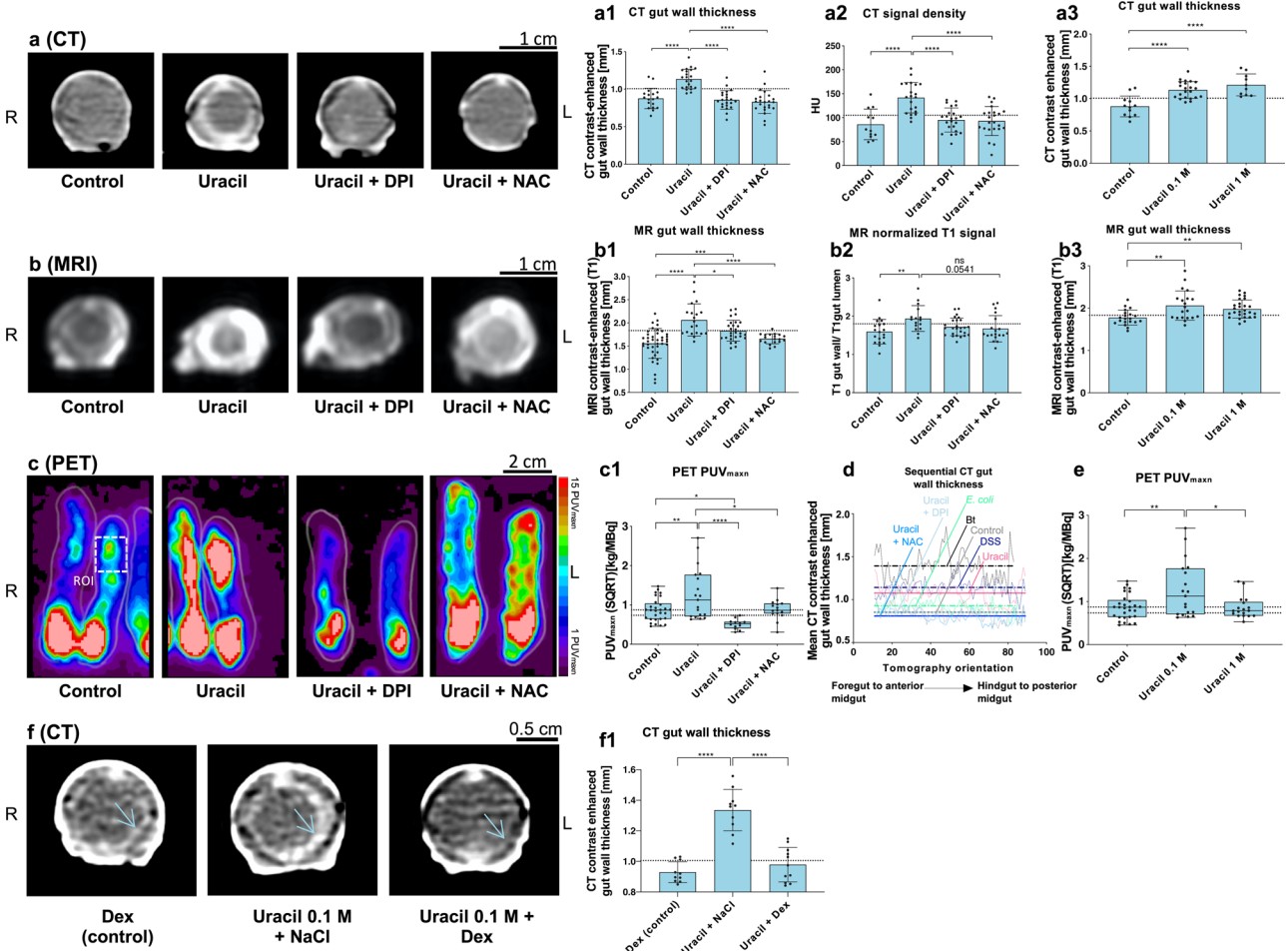

**Fig. 8 | High-throughput screening of an uracil-induced and DUOX-dependent colitis-like phenotype. a–c** High-throughput CT, MRI, and FDG-PET reveal an uracil-induced colitis-like phenotype, which is ameliorated by the DUOX inhibitor diphenyleneiodonium (DPI) or the ROS scavenger N-acetylcysteine (NAC), **a1:** $n = 85$, one-way ANOVA, $F(3,81) = 23.77$, $R^2 = 0.4682$, $P < 0.0001$, **a2:** $n = 79$, one-way ANOVA, $F(3,875) = 14.6$, $R^2 = 0.3687$, $P < 0.0001$, **b1:** $n = 107$, one-way ANOVA, $F(3,103) = 16.45$, $R^2 = 0.3239$, $P < 0.0001$, **b2:** $n = 80$, one-way ANOVA, $F(3,76) = 4.016$, $R^2 = 0.1368$, $P = 0.0104$ and **c1:** $n = 75$, one-way ANOVA, $F(3,71) = 9.627$, $R^2 = 0.2892$, $P < 0.0001$. **a3, b3,** and **e** Concentration-dependent effect of DUOX (activated by uracil) and successive worsening of the colitis-like phenotype **a3:** $n = 44$, one-way ANOVA, $F(2,41) = 16.48$, $R^2 = 0.4457$, $P < 0.0001$, **b3:** $n = 67$, one-way ANOVA, $F(2,64) = 6.809$, $R^2 = 0.1755$, $P = 0.0136$ and E: $n = 66$, one-

way ANOVA, $F(2,63) = 5.014$, $R^2 = 0.1373$, $P = 0.0095$. **d** Mean sequential CT gut wall thickness curves (the horizontal lines represent the overall treatment-specific mean thickness), preferred model: different curve for each data set, $F(6, 535) = 330.2$, $P < 0.0001$. **f, f1** CT imaging revealed the rescue of the uracil-induced phenotype by dexamethasone (Dex) treatment, $n = 31$, one-way ANOVA, $F(2,28) = 41.71$, $R^2 = 0.7487$, $P < 0.0001$. Dashed lines indicate threshold values. Significance levels: ns $= P > 0.05$, $*P \le 0.05$, $**P \le 0.01$, $***P \le 0.001$ and $****P \le 0.0001$. Bar charts represent mean and SD. Every data point represents a single animal. Box plots represent 25th–75th percentiles; whiskers represent min-max (show all points), and the centers represent median values. Source data are provided as a Source Data file.

contrast enhancement of the insect gut wall is assumed to be caused by the increased extracellular volume fraction, which in turn may reflect the destruction of septate junctions or cell separation, swelling, and/or death.

The potential range of applications for our screening system is very broad, as demonstrated by the analysis of host-pathogen interactions with resident *M. sexta* bacteria. Here, we were able to distinguish pathogens from intestinal mutualists and reveal a previously unknown protective function of the *M. sexta* gut microbiome[39]. This could easily be extended to bacteria relevant to human health[20]. We showed that DUOX shaped the microbiome of *M. sexta* towards a more HOCl-resistant composition (Fig. 9a, b and h, i). Furthermore, we demonstrated the close similarity of *M. sexta* and human DUOX, particularly within the PHD. In humans, DUOX is thought to be an integral component during the etiology of gut inflammation, but its exact role is unclear[47]. DUOX loss-of-function mutations are known in patients with very-early-onset inflammatory bowel disease (IBD)[48–50], and studies from both insects and mammals show increased susceptibility to

gastrointestinal infections after the knockdown of DUOX[11,26]. On the other hand, excess DUOX activity and subsequent ROS production can induce colitis-like patterns[47] as corroborated by our data. In line, elevated ROS production and subsequent impairments have been reported in IBD patients[51], and *DUOX* expression levels are much higher than normal in Crohn's disease (CD) and ulcerative colitis (UC) patients[52–54].

In this context, we further evaluated our screening platform by rescuing the uracil-induced phenotype with dexamethasone (Dex) treatment (Fig. 8f). Glucocorticoids like Dex are widely used as treatment of acute flares in CD and UC, suppressing several inflammatory pathways like the disruption of eicosanoid synthesis by inhibiting phospholipase $A_2$ (PLA$_2$)[55,56]. Intriguingly, also lepidopteran DUOX is controlled by PLA$_2$ activity and eicosanoid signaling[56]. This emphasizes the preclinical relevance of our method and further supports the DUOX-colitis-axis[57]. Unfortunately, appropriate mouse models are not available to investigate this relationship in more detail and up to now the role of DUOX in colitis or IBD has solely been analyzed with focus on its loss of function and not its overexpression or activation[11,27,58].

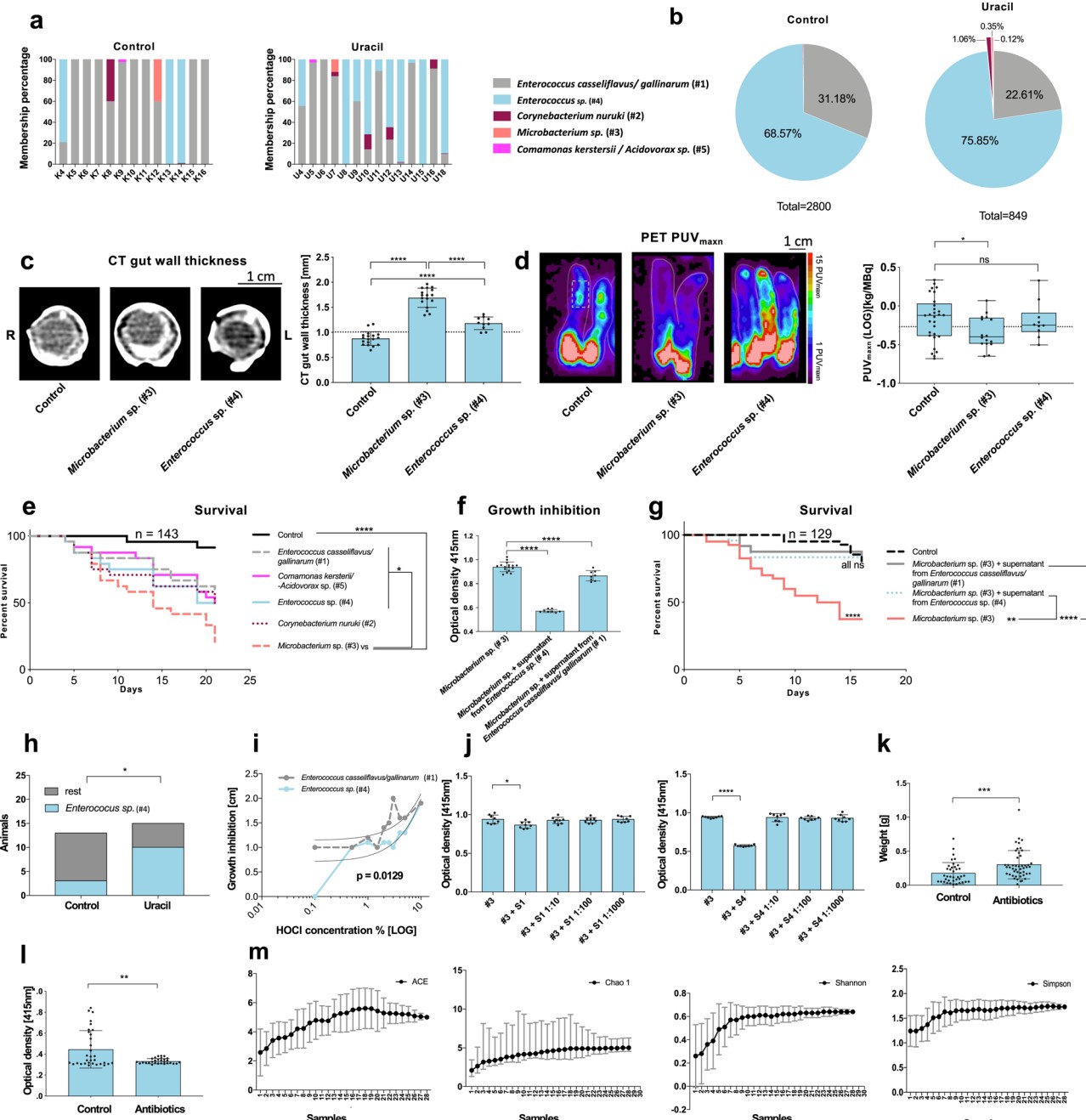

**Fig. 9 | Discrimination between intestinal bacterial pathogens and mutualists in *Manduca sexta* by multimodal imaging. a, b** The bacterial composition of fecal bacteria from control or uracil-fed animals (after 20.5 h). **c, d** Animals fed with the isolated pathogenic *Microbacterium* sp. (#3) differed significantly from control animals in terms of CT gut wall thickness ($n = 45$, one-way ANOVA, $F(2,42) = 117.5$, $R^2 = 0.8484$, $P < 0.0001$) and PET $PUV_{maxn}$ (two-tailed $t$ test, $P = 0.0437$, no adjustments), whereas symbiotic *Enterococcus* sp. (#4) differed only slightly in terms of CT gut wall thickness and not in PET $PUV_{maxn}$ (two-tailed $t$ test, $P = 0.8334$, no adjustments). Dashed lines indicate threshold values. **e** *Microbacterium* sp. is considered a pathogen because it reduced the survival of *M. sexta* significantly more than the other isolated bacteria. **f** Both *Enterococcus* spp. (containing a suppressive component against the *Microbacterium* sp.) suppressed the growth of the *Microbacterium* sp ($n = 32$, one-way ANOVA, $F(2,29) = 285.4$, $R^2 = 0.9516$, $P < 0.0001$). **g** Supernatant of the *Enterococcus* spp. rescued animals fed with *Microbacterium* sp. Therefore, the enterococci were defined as immunoprotective symbionts. **h** Although there was a total reduction in the CFU count after uracil treatment, significantly more *Enterococcus* sp. (#4) positive samples were found in animals treated with uracil, two-sided Chi-square-test, $P = 0.0211$. The reason for that finding could be the relative insensitivity of *Enterococcus* sp. #4 against HOCl

in comparison to *Enterococcus casseliflavus/gallinarum* (#1), **i** dose–response testing revealed an elevation difference: $F(1,17) = 7.711$. $P = 0.0129$. **j** Supernatant from both *Enterococcus* spp. inhibited the growth of *Microbacterium* sp. (#3), indicated through less measurable optical density, #3 + S1: $n = 40$, one-way ANOVA, $F(4,35) = 4.043$, $R^2 = 0.316$, $P = 0.0085$ and #3 + S4: $n = 40$, one-way ANOVA, $F(4,35) = 185.9$, $R^2 = 0.9551$, $P < 0.0001$. **k** Animals treated with penicillin grow significantly faster, indicating the pleiotropic effect of bacterial maintenance for *M. sexta*, two-tailed Mann Whitney test, $P = 0.0004$, no adjustments. **l** The clearance of gut bacteria through penicillin treatment was confirmed using fecal pellets' optical density measurements, two-tailed t-test, $P = 0.0013$ no adjustments. **m** Rarefaction curves showed an adequate sampling of the low-diversity bacterial community in fecal pellets of *M. sexta*. Cultivable bacteria isolated from the feces of *M. sexta* (see Tables S11 and S12). Significance levels: ns = $P > 0.05$, *$P \leq 0.05$, **$P \leq 0.01$, ***$P \leq 0.001$ and ****$P \leq 0.0001$. Survival kinetics show the sum of the conducted experiments. Bar charts represent mean and SD. Every data point represents a single animal. Box plots represent 25th-75th percentiles; whiskers represent min–max (show all points), and the centers represent median values. Source data are provided as a Source Data file.

In terms of translation, there are some structural differences between insect and mammalian DUOX, e.g., mouse and human DUOX1 structures are lacking the heme ligand[59,60] which may indicate the loss of peroxidase function in mammalian DUOX1 although the overall structure of the PHD remains highly conserved between all species. The only DUOX isoform present in the human gut is DUOX2[35] which exhibits significant peroxidase activity at least in human respiratory epithelium[61]. Functional effects are further supported by the finding that *Drosophila DUOX*-RNAi flies could be rescued by reintroduction of DUOX enzymes with human PHDs[26]. Interestingly, both known mutations of the DUOX2 PHD lead to lower $H_2O_2$ production in human IBD patients (p.P303R[48] and p.R286H[49]) further corroborating a potential role of the human PHD in bowel disease. Intriguingly, a binding site for N-acetylglucosamine (NAG) has been predicted in all modeled species, which could modulate the immune activity of DUOX (Tables S5–S8). Based on these findings, we propose DUOX as a promising antagonistic, pleiotropic target in the treatment of human gut inflammation and IBD, which should be pursued by the generation of genetically engineered mouse models with conditional overexpression of *DUOX2*.

In conclusion, our results confirm the feasibility of CT, MRI, and FDG-PET as persuasive high-throughput techniques to monitor the degree and progression of gut inflammation and infection in *M. sexta* larvae with OAI as potential additional modality. The simple geometry of the digestive tract in *M. sexta* larvae allows autonomous or semi-autonomous quantification, further improving the reproducibility of the platform and enabling the screening of large candidate libraries beyond traditional survival analysis. We have demonstrated that our approach can identify new therapeutic targets, monitor the success of novel pharmacological concepts, and assess host-symbiont interactions filling the gap between simple cell culture and complex mouse models[62–64]. The developed platform represents an ethically acceptable, resource-saving, large-scale and 3R-compatible screening tool for various life science disciplines, including (**i**) identification of new effectors and inhibitors in gut inflammation, (**ii**) assessment of pesticides or other environmental factors (**iii**), assessment and evaluation of new antibiotic therapies, (**iv**) analysis of host-pathogen interactions, and (**v**) identification of new contrast agents or tracers in radiology. Since 75% of the known human disease-causing genes have homologs in insects, this approach will also be helpful to test preclinical hypotheses (as demonstrated with DUOX) in other diseases, including cancer, diabetes, neurodegeneration, and infection[4,65,66].

# Methods

## Animal rearing preparation and ethics

The *Manduca sexta* larvae were reared at 24 °C with a 16 h photoperiod and were fed on a modified artificial diet[43] without preservatives (Details of the *M. sexta* diet are given in the source data). To establish imaging procedures, larvae were fed with artificial diet cuboids (7 × 7 × 4 mm) immersed in 0.3% (w/v) commercially available *Bacillus thuringiensis* (Bt) subsp. *aizawai* suspension (2.7 × 10^8 cells/ml) (Xentari, Neudorff, #00592), or containing 5% (w/v) dextran sodium sulfate (DSS, MP Biomedicals, #0216011025), *E. coli* (DH5α strain, New England Biolabs, Ipswich, MA, #C2987H) with 2.7 × 10^8 cells/ml (*E. coli* control), without any additions (control), or Bt with two concentrations of gentamicin (0.5 mg/ml or 1.0 mg/ml, Sigma-Aldrich, St. Louis, MO, #G1914) for 12 h.

For uracil experiments, the larvae were fed with food cuboids submerged in 0.1 M uracil (Carl Roth, Karlruhe, Germany, #7288.2) with or without DPI (65 μM, Cayman Chemical, Ann Arbor, MI, #81050) and NAC (72 μM, Carl Roth, Karlruhe, Germany, #4126.1) or with the normal diet (control) for 12 h. Further animals were fed on a regular diet containing 1.0 M uracil. One cuboid was used per animal. They were starved 1 h before feeding, and no additional food was applied. For the dexamethasone rescue experiment, larvae were fed as indicated above and injected with 100 μg dexamethasone (Sigma-Aldrich, St. Louis, MO,

#D4902) in 0.1 ml 0.9 % NaCl and exposed to a regular control diet. The inflammation group was exposed to 0.1 M uracil treatment as indicated above and injected with 0.1 ml 0.9% NaCl. The rescue group was exposed to 0.1 M uracil treatment and injected with 100 μg dexamethasone in 0.1 ml 0.9 % NaCl. CT imaging was done after 12 h of exposure.

For mutualist pathogen differentiation, L5 day 1 animals were fed on a regular diet (without preservatives) sprinkled with 250 μl of 2.7 × 10^9 cells/ml of the isolated *Microbacterium* sp. (#3) or *Enterococcus* sp. (#4) for 4 days. The food was renewed every day.

The detailed experimental procedure is shown in Fig. 10. Only animals of the same developmental stage (L5 larvae, day 5–6 if not indicated otherwise) were included in this study. We selected this developmental stage because the animals are very similar in size but also large enough for CT, MR, and PET (6.7 cm in length with a coefficient of variation = 9.3%; Fig. 10). Abnormal animals or animals with morphological or behavioral signs of late L5 development (e.g., a noticeable dorsal vessel or cessation of feeding) were excluded.

Unlike vertebrate research animals, working with insects does not require any approval in Europe or the United States.

## Magnetic resonance imaging

The Bt-infected larvae were imaged at 12, 36, and 42 h post-feeding, while all other treatments were imaged 12 h post-feeding (Fig. 10). For immobilization, larvae were cooled for 30 min on ice and then injected into the dorsal vessel (at the level of the 7th abdominal segment) with 0.1 ml 0.2 M Gd-BOPTA (MultiHance, Bracco Diagnostics, Milan, Italy, #12406641) in 0.9% NaCl or fed with Gd-BOPTA for 12 h. For further immobilization, animals were carefully fixed with Leukosilk S (BSN Medical, Charlotte, NC) on cardboard. The isolated midgut was exposed to 0.2 M Gd-BOPTA in 0.9% NaCl for 5 min, and then washed twice with 0.9% NaCl and deposited in 0.9% NaCl for MRI. Images were taken 15 min after the application of contrast agent (CA) with a standard four-channel flex coil in a clinical Siemens Magnetom Symphony 1.5 T MRI system with a maximum gradient field strength of up to 30 mT/m (Siemens Healthineers, Erlangen, Germany). Axial T1-weighted images were measured using an SE sequence (+FS) with the following settings: RT/TE = 736 ms/15 ms, flip angle = 90°, FOV = 178 × 260 mm, matrix = 512 × 352, and a slice thickness of 3 mm. Coronal T1-weighted SE sequences with FS were acquired with the following parameters: RT/TE = 451 ms/15 ms, flip angle = 90°, FOV = 280 × 280 mm, matrix = 512 × 512, and a slice thickness of 3 mm. Furthermore, axial T2-weighted images were recorded using an SE sequence with the following settings: RT/TE = 4720 ms/96 ms, flip angle = 180°, FOV = 280 × 512 mm, matrix = 512 × 208, and a slice thickness of 3 mm. Coronal T2 weighted SE images were measured with the following settings: RT/TE 2000 ms/96 ms, flip angle = 180°, FOV = 220 × 220 mm, matrix = 512 × 261, and a slice thickness of 3 mm. The T1-weighted axial sequences were analyzed with Horos v3.3.5. The signal enhancement in T1-weighted sequences was calculated and defined as the normalized T1 signal. The ratio of the maximum T1-weighted signal from the gut wall to the T1-weighted signal from the gut lumen was calculated for every slice. From these values, mean values for each animal were calculated. The maximum gut wall thickness measured in contrast-enhanced T1-weighted MRI was measured manually at the thickest visible spot on each slice. The accuracy of these measurements was verified through a Full Width at Half Maximum (FWHM) analysis (Figs. S7, S8). Then mean values were calculated as described above. The T2-weighted signal of the gut wall was measured as the maximum T2 signal value on every slice, and mean values per animal were calculated.

## Computed tomography

All CT scans except the dexamethasone rescue experiment were performed on a clinical Siemens SOMATOM Emotion 6 (110 kV, 80 mAs, and 1 mm collimation) (Siemens Healthineers, Erlangen, Germany).

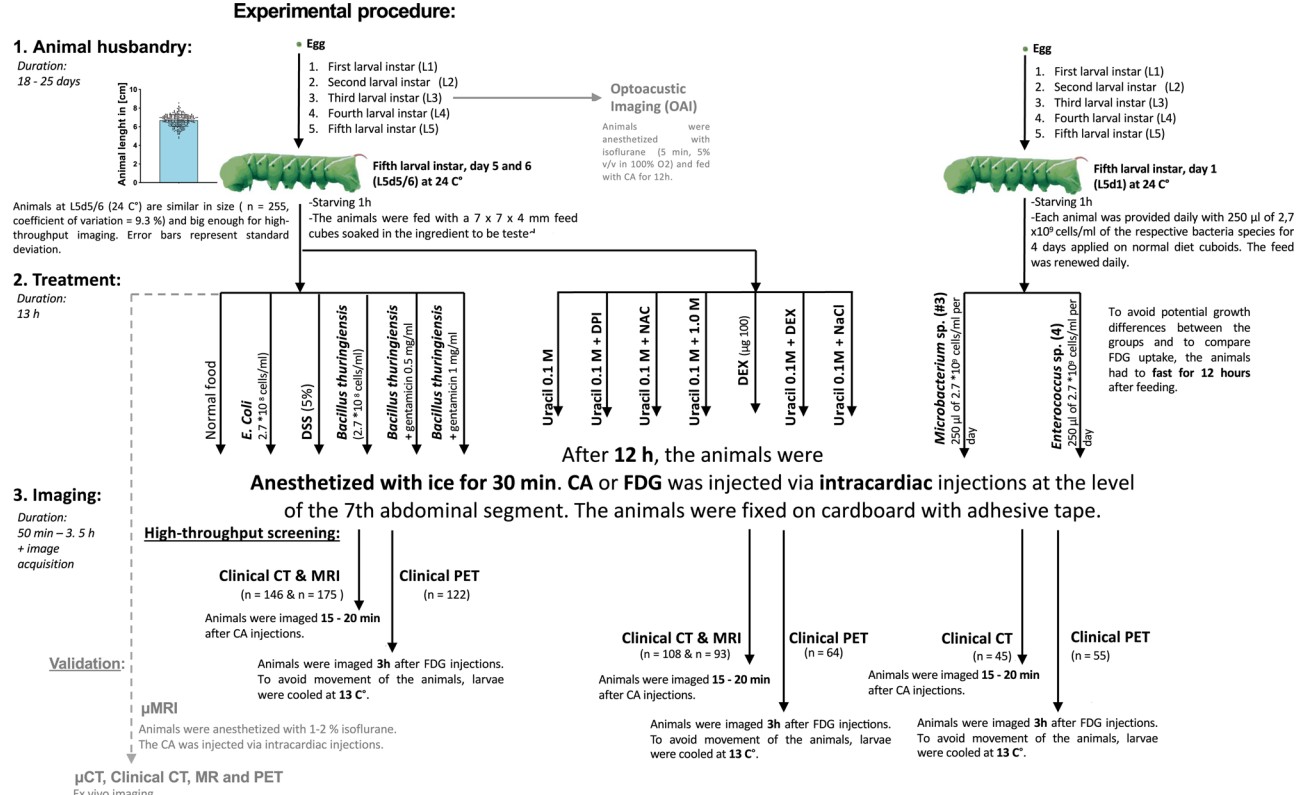

**Fig. 10 | Experimental procedure of CT, MRI, and PET imaging used in this study.** Bar charts represent mean and SD. Every data point represents a single animal. Source data are provided as a Source Data file.

Larvae were prepared as described for the MRI experiments and injected or fed with 0.1 µl of 50% iodixanol (Visipaque 320, GE Healthcare, Solingen, Germany, #1105030) in 0.9% NaCl (Fig. 10). Time-dependent differences in the contrast-enhanced gut wall thickness of control and Bt-infected animals were measured at 9, 14, and 19 min. The isolated larval midgut was exposed to 50% iodixanol in 0.9% NaCl for 5 min, then washed twice with 0.9% NaCl and deposited in 0.9% NaCl for CT imaging. After CT, axial sequences were analyzed with Horos v3.3.5. The reconstruction interval was 1.25 mm. The maximal gut wall thickness in contrast-enhanced CT was measured as described for MRI, and the maximal contrast-enhanced CT gut wall signal density was measured at maximum density on each slice, with mean values calculated for each animal. A subset of animals without gaps in the gut wall thickness measurements along the whole midgut was selected for sequential treatment-specific analysis of the axial CT scans (Figs. 4d and 8d). For better comparability of the thickness measurements, the first 10 measurements were excluded. The resulting treatment-specific sequential gut wall thickness curves (n : c = 12, u = 22, u+DPI = 22, u+NAC = 23, Bt=13, DSS = 14, *E.coli* = 12) were compared with an extra sum-of-squares F test and tested whether each treatment had the same slope and intercept and could be represented as a single global model (H0) or if individual models for each treatment should be employed (H1).

The dexamethasone rescue experiment was imaged with a SOMATOM Force (Siemens Healthineers, Erlangen, Germany) with the following settings: 110 kV, 102 mAs, and 0.6 mm collimation. All other settings were kept as close as possible to the SOMATOM Emotion 6 scans.

## Positron emission tomography
We followed the same protocol as described above for MRI (Fig. 10). Larvae were fed with Bt, DSS, *E. coli*, or control diets and fasted for 12 h. Cooled and immobilized animals were injected with 0.1 ml 1 MBq/ml 2-[18F]-deoxy-D-glucose (FDG, Life Radiopharma f-con, Holzhausen, Germany, e.g., #220107.1) in 0.9% NaCl in the caudal dorsal vessel and kept cool at 13 °C. FDG PET was performed 3 h after injection in larvae 3, 12, and 24 h after Bt infection with a PEM FLEX Solo II system (CMR Naviscan, Carlsbad, CA) using an in-plane spatial resolution of 1.8 mm and a between-plane resolution of 4–6 mm as previously described[67]. The emission scan time was 15 min. Maximum PUV ($PUV_{max}$) was measured for each larva in a rectangular region enclosing the apical third of each animal without the head using MIM Viewer (PEM b1.2.4; Fig. 1o). To ensure high-throughput PET screening, we used 1 kg and 1 MBq as the default option to measure $PUV_{max}$. The $PUV_{max}$ was normalized for the activity and mean weight of the larvae as the $PUV_{maxn}$. We determined the $\bar{x}$ mean weight of the larvae, which were very similar in size, as 7.419 g (Fig. 10).

The $PUV_{maxn}$ was calculated using the following formula:

$$PUV_{maxn} = \frac{PUV_{max}}{\frac{\text{activity [MBq]}}{\bar{x}\text{ weight [Kg]}}} \tag{1}$$

We also isolated the midgut ($\bar{x} = 0.81$ g), the head ($\bar{x} = 0.296$ g), as well as fat body tissue and hemolymph (both weighed individually because uniform sampling was not possible), and compared the $PUV_{maxn}$ values 39 min or 3 h after 18F-FDG injection in animals 12 h after Bt infection. The removed tissue was washed in 0.9% NaCl.

## Statistics
For statistical analysis, we generally used PRISM v8, with only the general linear model calculated using Statistica v12.5.192.7. To evaluate different parameters for CT and MRI diagnostics, all axial slices from a given larvae were measured, and the mean value per animal was calculated. $PUV_{maxn}$ values were log or square root transformed to achieve normal distribution. Depending on the data distribution (evaluated using the Shapiro-Wilk test), we used one-way analysis of variance

(ANOVA) and Tukey's multiple comparisons test or the Kruskal-Wallis test and Dunn's multiple comparisons test. When two treatments were compared, we applied a t-test or Mann–Whitney U-test (two-sided). A Pearson product-moment correlation between parameters was calculated to identify correlations between the CT, MRI, and PET findings. Images were inspected and screened for artifacts directly after image acquisition. Larvae were excluded if a large amount of CA spilled over after injection into the dorsal vessel due to animal movement during image acquisition (MRI control = 2, DSS 5% = 2; CT DSS 5% = 1, E. coli = 1). Additionally, three outliers were excluded using the ROUT ($Q = 1\%$) method (MRI control = 1; PET DSS 5% = 2). To test whether small potential differences in larval size affect the epithelial thickness, we used general linear models with MRI/CT gut wall thickness or PET $PUV_{maxn}$ as the dependent variable, treatment as a category factor, and animal length as a continuous predictor. ROC curve analysis was carried out using PRISM, including sensitivity and specificity for the listed parameters. The corresponding threshold values were used to compare Bt-infected, DSS, or E. coli-fed, and control animals using multiple chi-square tests. The manual thickness measurements were validated via semi-automatic FWHM thickness measurements using OriginPro 2020b and Analyze 14.0. Bland Altman plots have been created using PRISM.

## Study approval
Unlike vertebrate research animals, working with insects does not require any approval in Europe or the United States.

## Reporting summary
Further information on research design is available in the Nature Portfolio Reporting Summary linked to this article.

# Data availability
All data supporting the findings of this study are available within the article and its supplementary material. All DNA sequencing data were uploaded to GenBank sequence database (OP630947-OP630951). Raw imaging data are available from the corresponding authors. Source data are provided with this paper.

# Code availability
Not applicable.

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

## Acknowledgements

We are grateful to Jessica Steinbart (µCT and CT) and the team from Radiologie und Nuklearmedizin Ludwigshafen for their technical support in image acquisition. Furthermore, we thank Dr. Sebastian Faby (Siemens Healthineers, Erlangen, Germany) for his help in imaging the Derenzo phantom in different CT scanners. We also thank Prof. Dr. Michael Kanost (Kansas State University), Prof. Maria Grandoch (Heinrich Heine University), and PD Dr. Ellen Kauschke (Justus Liebig University Giessen) for their helpful comments on the manuscript. Finally, we thank Carolin Ina Schröter for the artwork of *M. sexta* and the bacteria. This work was supported by the Deutsche Forschungsgemeinschaft (INST 208/764-1 FUGG to U.F.) and the NIH/NCI (P30 CA008748 Cancer Center Support Grant to J.G.).

## Author contributions

A.G.W., T.E.T., and U.F. conceived of the main concept and guided the project; F.H.H.M., B.M., and J.G. contributed to the study design; A.G.W., F.H.H.M., B.M. M.H., A.C.B., M.K., V.F., and U.F. performed experiments and/or analyzed the data; A.G.W., C.R.B., and F.H.L. carried out histopathological analysis; L.M. provided materials; Y.B. contributed to project administration, H.M., G.A.K., A.V., and J.G. helped with critical advice and discussion. A.G.W., T.E.T., and U.F. drafted the manuscript and all authors discussed the results and contributed to the writing of the paper.

## Funding

## Competing interests

The authors declare no competing interests.

## Additional information

[1]Institute of Zoology and Developmental Biology; Cellular Recognition and Defense Processes, Justus Liebig University Giessen, Giessen, Germany. [2]Department of Bioresources, Fraunhofer Institute for Molecular Biology and Applied Ecology IME, Giessen, Germany. [3]Laboratory of Experimental Radiology, Justus Liebig University Giessen, Giessen, Germany. [4]Radiology and Nuclear Medicine Ludwigshafen, Ludwigshafen, Germany. [5]Molecular Pharmacology Program, Memorial Sloan Kettering Cancer Center, New York, NY, USA. [6]Molecular Imaging and Therapy Service, Memorial Sloan Kettering Cancer Center, New York, NY, USA. [7]Department of Nuclear Medicine, Inselspital Bern, Bern, Switzerland. [8]Department of Chemistry and Biology, School of Science and Technology, University of Siegen, Siegen, Germany. [9]Applied Zoology, Department of Biology, Technical University of Dresden, Dresden, Germany. [10]Experimental Cardiovascular Imaging, Molecular Cardiology, Heinrich Heine University Düsseldorf, Düsseldorf, Germany. [11]Department of Diagnostic and Interventional Radiology, University-Hospital Giessen, Giessen, Germany. [12]Institute for Insect Biotechnology, Department of Applied Entomology, Justus Liebig University Giessen, Giessen, Germany. [13]Pharmacology Department, Weill Cornell Medical College, New York, NY, USA. [14]Department of Radiology, Memorial Sloan Kettering Cancer Center, New York, NY, USA. [15]Department of Radiology, Weill Cornell Medical Center, New York, NY, USA. [16]These authors jointly supervised this work: Tina E. Trenczek, Ulrich Flögel. ✉e-mail: Tina.Trenczek@gmx.de; floegel@uni-duesseldorf.de

