## [Peer Review File · Nature Communications]

REVIEWER COMMENTS

Reviewers' expertise:

Reviewer #1. Insect physiology and microbiome.

Reviewer #2. Insect physiology and microbiome.

Reviewer #3. CT-based imaging approaches in insects

Reviewer #4. Chemical biology, high throughput screening and PET imaging.

Reviewer #1 (Remarks to the Author):

This study establishes a high-throughput method to experimentally induce colitis in the tobacco hornworm (*Manduca sexta*) and analyze its pathogenesis using CT and PET, as well as a detailed analysis of Duox-mediated colitis in the caterpillars, thereby presenting the possibility that the caterpillars and its analysis system can serve as a useful platform for exploring treatments for colitis and effective microbes. Specifically, the authors showed that the severe colitis experimentally induced by BT and surfactants in caterpillars can be detected in considerable detail by CT and PET in a high-throughput manner. The biological function of Duox in gut immunity was also analyzed by similar methods, and the dynamics of the gut microbiota during this process were also observed.

The study is very unique for an entomologist such as me and is conducted using an extremely wide range of high techniques, making it catchy and easy to understand. In addition, this study is well organized and written with a lot of data. However, considering that the purpose of this study is to give a better (i.e. more high-throughput and cheaper) platform for drug discovery, I still have some concerns about the benefit of this platform.

It would be possible to screen drugs or microorganisms that would prevent the colitis-like symptoms in the caterpillar; however, at least from the data shown in this paper, it is still quite unclear whether such drugs or microbes would perform the same function in humans or mice in the first place (given that BT is harmless to humans, it is highly unlikely that it would be practical to screen drugs or microbes that prevent BT as human drugs). If the authors want to argue this point convincingly, they need to show some valid data. For instance, data showing that some antifatulent or colitis drugs that have already been proven in humans and mice are actually effective in improving the colitis-like symptoms in the caterpillars would give a strong impression that this system is actually useful for drug discovery. Such additional data is pivotal to proving the benefit of this platform.

Duox has also investigated in great detail, whose results were well described and interpreted. On the other hand, despite the great amount of data devoted to it, this study does not contain any new insights into the biological function of Duox. Instead, the previous findings that Duox contributes to the development of peritrophic membrane (PM) and plays a pivotal role in the trachea network surrounding the gut are completely ignored. It should be clarified the relationship between the phenomenon described in this study and the pleiotropic effect of Duox in the gut.

Although the platform claims to be a platform for analyzing gut immunity, only Duox has been investigated. Although the authors stated “innate immunity” in the title, more important molecules such as antimicrobial peptides (AMPs) were not included in this study. As shown in *Drosophila* and mosquitoes, AMPs are regulated by transcription factors such as caudal, or bacterial components that stimulate immune cells (fat bodies) to release AMPs into the gut tract when PM is disrupted (e.g. by loss of function of Duox). It remains unclear to what extent the major gut immunity can be studied on this platform.

Other comments

L301 – 303 “Electron microscopy revealed the loss of,,, but not the controls (Fig. 3C+D and S9).”: Are there any changes in the peritrophic membrane?

L343 – 345: It remains unclear how they correspond to the symptoms of Crohn's disease.

L387 – 388 (Fig. 9F-I): Why are figure 9F-I not involved in Figure 7?

L424 – 428 “Overall, these findings demonstrate that our approach is,,, testing and high-throughput screening of new therapeutic concepts.”: But, as the authors are also discussing, it remains under investigation whether DUOX is involved in IBD.

L527 – 543 A possible role of DUOX in IBD: Although this prediction is interesting, the hypothesis should be confirmed by experiments.

Reviewer #2 (Remarks to the Author):

In this manuscript, the authors applied diagnostic imaging technologies to hornworm larvae as tools to investigate intestinal pathologies caused by infection, commensal dysbiosis, and colitis-inducing agents. They propose to use these imaging technologies of insect larvae as a high-throughput

screening platform for the identification of new effectors and inhibitors of gut inflammation, new antimicrobials, and study of host-pathogen interactions among other applications. The authors used 3 cases to illustrate the functionality of their approach: Bt infection, uracil treatment, and infection with a native member of hornworm flora. Overall, this is a well-performed study with an enormous amount of data. It establishes a new high-throughput screening platform in insects that can be used for a variety of purposes and is of potential interest to researchers in several fields.

I have a few suggestions that can help to improve the manuscript.

1. This is a very data-rich manuscript, however many of the results are not discussed at all or just mentioned very lightly. The authors should make an effort to better explain their results and make sure that every panel of each figure is mentioned in the text (this is not the case now). For example, I would appreciate an explanation how FDG is used to evaluate metabolism (line 236).

Also, lines 351-352, this sentence needs a better explanation.

2. The authors should organize the figures in a more consequential order or reorganize the text in a way that the readers do not need to go back and forth with the figures.

3. In line 397, the authors refer to DPI as bona fide Nox inhibitor but used it to inhibit Duox. Is there any evidence that DPI inhibits Duox? How do the authors know that the effect that they observed with DPI is due to Duox and Nox inhibition? It would be nice to provide an experimental confirmation that Duox is inhibited by DPI in *M. sexta*.

4. The authors should provide more details on the data behind graphs. Since they keep referring to high-throughput but never mention how many animals were analysed it is not clear how high-throughput it is. What do bar graphs show, e.g. mean+SD? What does each dot mean? How many animals were used? How many repeats? For survival graphs, what do they show: representative experiment? average of n experiments? How many animals were used per treatment?

5. Western blot in Fig. 7B, 7C. Was a specific gut region taken for quantification or all regions were combined? Explain in the legend. Since there is no loading control, how the authors can be sure that the differences are not due to differences in the loaded amount?

6. Fig. 4H lacks statistical analysis.

Line 195. Toll is also an NF- κ B pathway, did the authors mean Relish?

Reviewer #3 (Remarks to the Author):

Overall Assessment

The manuscript by Windfelder et al. provides an in-depth analysis regarding the suitability of using caterpillars as a model for gut inflammation that might be relevant to human health. The authors show that preclinical imaging modalities, particularly CT, can serve as a rapid diagnostic tool for gut health in an insect system, which may serve as a suitable alternative to traditionally used mammalian models. They go on to validate their model and to show further similarity at the molecular level between insect and mammalian (particularly humans) proteins known to be important for modulating gut homeostasis (DUOX) and the response to bacterial pathogens. Overall, the amount of data included in this paper is impressive, and while this reviewer would have appreciated perhaps a more streamlined approach to the resulting figures and supplementary material, I do appreciate the quality of the work and the tone of the conclusions that are drawn from them. While some mammalian folks may not recognize the significance of these results, I do believe that the authors have made a good case for using *M. sexta* as a model system, especially if the major concerns that I have outlined below can be addressed.

Major concerns

1. What is the actual spatial resolution of the CT images? The Voxel sizes reported here (350 μ m for CT, 7.55 μ m for u-CT) describe the properties of the instrument (detector size, geometry) and do not accurately reflect the actual spatial resolution of the images. Actual resolution in CT & u-CT can be influenced by many different factors, and may be slightly different from one scan to the next even with identical imaging parameters. I would advise the authors to use a phantom designed to determine actual spatial resolution at the imaging conditions used. The phantoms provided in Fig. S4 do not satisfy this criteria, as they only show that the resolution is sufficient to \sim 4 mm.

2. The actual spatial resolution is important because it can directly impact the accuracy of the thickness measurements. For example, significant differences in gut thickness is reported (in Fig. 1M, 4A1, 4D, 4F, for example) that vary by \sim 250-400 microns. Assuming that the actual spatial resolution of the CT scans are \sim 1 millimeter, this puts the actual size of the gut walls themselves either at or

below the resolution limit. At the minimum, this could directly impact the ability to detect less severe (but nonetheless biomedically relevant) forms of gut inflammation and should be addressed accordingly in the manuscript.

3. It is also unclear to me how the thickness measurements were made for CT and MRI. The Methods state “The maximum gut wall thickness measured in contrast-enhanced T1-weighted MRI was measured at the thickest visible spot on each slice”; with the CT measurements “as described for MRI”. Were the measurements done by manually drawing a line across the gut? How were boundaries for these determined—was a certain pixel intensity threshold used? This could be highly error prone, especially if the resolution of the tomograms are not sufficiently high (a single pixel could therefore throw off a measurement by 100 microns or more). Please clarify this in the Methods section.

4. Considering the issues outlined above, it might be preferable to perform a Full Width Half Maximum (FWHM) analysis of line scans across the gut wall for the width measurements in order to verify the accuracy of these measurements, and also lessen the concern about the resolution limits imposed by CT and the size of the gut wall. FWHM measurements do not rely on manual determination of signal boundaries and instead rely on mathematical Gaussian fits to the line scan data. Also, FWHM measurements have become the standard approach for measuring optically unresolved structures in fluorescent light microscopy (standard vs. super resolution, for example) to provide accurate and unbiased parameters for resolution comparisons below the diffraction limit.

Specific Comments

Pg 6., lines 1-14: I think this argument could be further strengthened by incorporating a statement on the availability/use of genetic tools in *M. sexta* to help elucidate molecular mechanisms.

Pg 8, lines 3-5: The sample size for micro-CT measurements are low. A higher n would give further confidence in the CT image measurements despite the potential spatial resolution issue outlined above.

Would the inability to detect a significant increase in CT signal density for DSS treatment (Fig. 4A2) despite a significant increase in gut wall thickness limit the applicability of CT for mild models of gut inflammation?

In an effort to streamline the main figures, I would suggest moving Fig. 5 to the supplement, at least 5A-F.

Pg. 19, line 3: later should be latter.

Pg. 24, lines 9 & 11. Italicize *M. sexta*

Pg. 25, Figure legend: Remove uracil from title, as this figure is just for BT treatment?

Pg. 37. Consider moving Fig. 10 to supplement

Pgs. 39-40. It is not clear to me exactly how the thickness measurements were made? And what exactly were the 3 different curve fitting tests that were used to identify differences in the CT gut wall thickness curves?

Fig S3 legend: Plains=Planes

Reviewer #4 (Remarks to the Author):

- What are the noteworthy results?

They have tested and validated a model non vertebrate organism for screening innate immune phenotypes by both anatomical and molecular imaging. This fills a gap between fruit flies and nematodes which can be examined by microscopic techniques and rodents that can be molecularly phenotyped by structural and molecular imaging.

- Will the work be of significance to the field and related fields? How does it compare to the established literature? If the work is not original, please provide relevant references.

Identifying phenotypes that are consistent across size and evolutionary time between this new organism and mice might improve the translational leap between mice and humans. An interesting question to be determined over time.

- Does the work support the conclusions and claims, or is additional evidence needed?

- Are there any flaws in the data analysis, interpretation and conclusions? - Do these prohibit publication or require revision?

While the goal is high throughput screening the analysis of imaging data is primarily focused on groupwise difference comparison. This may be a useful initial step. Analysis of effect sizes, eg z' or at least large cohens d effect sizes are more useful. As would be a deeper dive into test retest analysis and limits of agreement. Without this deeper dive there central thesis of high throughput screening remains to be validated.

- Is the methodology sound? Does the work meet the expected standards in your field?

See above. Experiments are indeed interesting and informative and interdisciplinary but the hypothesis to be tested and null hypothesis to be rejected do not drive to the "pitch". This can be corrected without additional experiments but must be corrected prior to publication.

- Is there enough detail provided in the methods for the work to be reproduced?

Yes

RESPONSE LETTER

We thank all reviewers for carefully evaluating our manuscript and for their constructive criticism. Their suggestions and comments have helped to improve our manuscript significantly. We have addressed all issues point by point, as outlined below.

As suggested by the reviewers, we carried out a variety of new experiments resulting in a large amount of new data (**Figures 2C and D, 4H, 7C, 8F and 10, Supplemental Figures S5, S6, S7, S8, S9, S22, S23 and table 14, 15, 16 and 17**) and major modifications of the manuscript. Furthermore, we provide a marked version of the manuscript where all alterations are highlighted in yellow. Note that display items that exclusively appear in the response letter are numbered with roman numbers.

Response to Reviewer #1

Overall Assessment:

This study establishes a high-throughput method to experimentally induce colitis in the tobacco hornworm (*Manduca sexta*) and analyze its pathogenesis using CT and PET, as well as a detailed analysis of Duox-mediated colitis in the caterpillars, thereby presenting the possibility that the caterpillars and its analysis system can serve as a useful platform for exploring treatments for colitis and effective microbes. Specifically, the authors showed that the severe colitis experimentally induced by BT and surfactants in caterpillars can be detected in considerable detail by CT and PET in a high-throughput manner. The biological function of Duox in gut immunity was also analyzed by similar methods, and the dynamics of the gut microbiota during this process were also observed. The study is very unique for an entomologist such as me and is conducted using an extremely wide range of high techniques, making it catchy and easy to understand. In addition, this study is well organized and written with a lot of data

Thank you for this positive evaluation!

Main comments:

However, considering that the purpose of this study is to give a better (i.e. more high-throughput and cheaper) platform for drug discovery, I still have some concerns about the benefit of this platform.

We agree with the reviewer's comment between the lines that the imaging platform used is not a cost-effective solution. However, it allows quantitative *in vivo* high-throughput approaches beyond simple survival kinetics and provides considerably more information.

Please note that we have used commonly available clinical imaging equipment (like CT, MRI, and PET) for the main experiments and used specialized small animal equipment only for validation experiments. Furthermore, we have conducted our experiments in an outpatient office and a hospital, mainly on the weekends and in the evening, when the scanners were not used. Thus, only low costs arose, and patient allocation problems were avoided. Importantly, existing high-throughput screening approaches with insects mostly use animal survival as a readout, but these strategies can only detect gross effects. However, other approaches capable to be more discriminative using techniques like PCR, WB, or immunohistochemistry are not applicable for larger sample sizes.

We have modified the manuscript to address this points as follows.

Page 5, line 156-157: *To this end, we used commonly available standard clinical scanners and small animal equipment only for validation studies.*

It would be possible to screen drugs or microorganisms that would prevent the colitis-like symptoms in the caterpillar; however, at least from the data shown in this paper, it is still quite unclear whether such drugs or microbes would perform the same function in humans or mice in the first place (given that BT is harmless to humans, it is highly unlikely that it would be practical to screen drugs or microbes that prevent BT as human drugs). If the authors want to argue this point convincingly, they need to show some valid data. For instance, data showing that some antifatulent or colitis drugs that have already been proven in humans and mice are actually effective in improving the colitis-like symptoms in the caterpillars would give a strong impression that this system is actually useful for drug discovery. Such additional data is pivotal to proving the benefit of this platform.

This point is well taken and we carried out additional experiments to address these suggestions. Using microorganisms as an alternative IBD treatment is an exciting new approach currently under investigation but with mixed results. Therefore, we decided to design an experiment using the anti-inflammatory drug dexamethason that is routinely used in the treatment of colitis ulcerosa (CU) and Crohn's disease (CD)^{1,2}. Insects and Lepidopteran are well known to be responsive to the treatment with anti-inflammatory corticosteroids³⁻⁷. Also, corticosteroids are a common choice in treating moderate or severe relapses of IBD^{1,2}. Thus, we expanded our previous uracil challenge experiments by additional treatment with dexamethasone, employed CT to measure the gut wall thickness of all animals and were thereby able to demonstrate the success of this therapy. These new results were now included as **Fig. 8F** into the revised manuscript (see also below).

Fig. 8: High-throughput screening of an uracil-induced and DUOX-dependent colitis-like phenotype (F–F1) CT imaging revealed the rescue of the uracil-induced phenotype by dexamethasone (Dex) treatment.

Page 39, line 851-857: *For the dexamethasone rescue experiment, larvae were fed as indicated above and injected with 100 µg dexamethasone (Sigma-Aldrich, St. Louis, MO) in 0.1 ml 0.9 % NaCl and exposed to a regular control diet. The inflammation group was exposed to 0.1 M uracil diet as indicated above and injected with 0.1 ml 0.9 % NaCl. The rescue group was exposed to 0.1 M uracil diet and injected with 100 µg dexamethasone (Sigma-Aldrich, St. Louis, MO) in 0.1 ml 0.9 % NaCl. CT imaging was carried out 12 h after exposure to verify the success of this therapy.*

Page 13-14, line 368-370: *Finally, dexamethasone treatment prevented the uracil-induced colitis-like phenotype in *M. sexta*, further corroborating the screening value of our platform (Fig. 8F+F1).*

Page 18, line 484-490: *In this context, we further evaluated our screening platform by rescuing the uracil-induced phenotype with dexamethasone (Dex) treatment (Fig. 8F). Glucocorticoids like Dex are widely used as treatment of acute flares in CD and UC, suppressing several inflammatory pathways like the disruption of eicosanoid synthesis by inhibiting phospholipase A2 (PLA2)^{55,56}. Intriguingly, also Lepidopteran DUOX is controlled by PLA2 activity and eicosanoid signaling⁵⁶. This emphasizes the preclinical relevance of our method and further supports the DUOX-colitis-axis⁵⁷.*

Duox has also investigated in great detail, whose results were well described and interpreted. On the other hand, despite the great amount of data devoted to it, this study does not contain any new insights into the biological function of Duox. Instead, the previous findings that Duox contributes to the development of peritrophic membrane

(PM) and plays a pivotal role in the trachea network surrounding the gut are completely ignored. It should be clarified the relationship between the phenomenon described in this study and the pleiotropic effect of Duox in the gut.

Thank you for making us aware of the pivotal role of DUOX in the trachea network and its contribution to the development of the peritrophic membrane. Indeed, we have characterized DUOX as a pleiotropic antagonistic target but focused solely on the bactericidal features and its role in autoimmunity. However, as this reviewer points out, DUOX is also crucial for tyrosine cross-linkage, which is essential for the integrity of the peritrophic membrane and the tracheal network, as Kumar *et al.* 2010 and Jang *et al.* 2021 have shown^{8,9}.

To address the valid comment of this reviewer, we have modified the manuscript and mentioned the findings from Kumar *et al.* 2010 and Jang *et al.* 2021 as follows:

Page 12, line 322-323: *Beyond its physiological importance in tyrosine cross-linkage, DUOX plays an essential role in the mucosal immunity of the gut*^{8,9}.

Although the platform claims to be a platform for analyzing gut immunity, only Duox has been investigated. Although the authors stated “innate immunity” in the title, more important molecules such as antimicrobial peptides (AMPs) were not included in this study. As shown in *Drosophila* and mosquitoes, AMPs are regulated by transcription factors such as caudal, or bacterial components that stimulate immune cells (fat bodies) to release AMPs into the gut tract when PM is disrupted (e.g. by loss of function of Duox). It remains unclear to what extent the major gut immunity can be studied on this platform.

We have taken up the reviewer's comments, conducted additional experiments and included an analysis of AMPs (Attacin 1 and Gloverin, **Fig. S15F**, see Figure next page and also below) after the infection with Bt to further confirm the colitis-like phenotype as observed in CT, MR, and PET.

Moreover, we agree that the innate immunity of the insect gut is indeed a large field. We have decided to include this into the title of the manuscript because we feel that our work may give new impulses to the scientific community studying insect immunity. Importantly, the methods we introduced are not limited to the analysis of DUOX. Beyond this, these advanced techniques align with the existing methods in insect innate immunity (like PCR, WB, and IHC). Further targets that might be involved in mucosal autoimmunity such as NOS, NOX, catalase, and GPx can be analyzed by our approach as almost any process resulting in tissue destruction of the gut and other tissues. This includes also – as we have shown with Bt and *Microbacterium sp.* – infections and the analysis of almost any inflammation or processes that promote/prevent the latter (including the analysis of AMPs and immune pathways like imd or DUOX), which is at the foundation of insect immunology.

In addition, we think that other promising aspects of these clinical imaging modalities will enhance insect immunology once they are more established for this. Here, special focus may be on tracing hemocytes, analysis of phagocytosis or polyphenol oxidase activity with PET tracers.

Page 8-9, line 245-246: *We also observed the induction of AMP genes encoding gloverin and attacin-1 in the midgut 24 h after Bt infection (Fig. S15F).*

Fig. S15: Bacteria found in the hemolymph, melanization of the gut, and induction of AMPs after Bt treatment

(F) Semiquantitative PCR analysis of AMPs in the medial midgut of *M. sexta* larvae after Bt infection. Lane c 1–3 show cDNA samples from the medial midgut of control animals (n=3), and lane Bt 1–3 show samples from animals treated with Bt (n=3). *EF1(α)* was used as a control.

Specific comments

L301 – 303 “Electron microscopy revealed the loss of ... but not the controls (Fig. 3C+D and S9).”: Are there any changes in the peritrophic membrane?

That is an interesting question. However, due to technical reasons we had to remove the peritrophic membrane in order to access the microvilli brush border via EM. Therefore, we unfortunately cannot address this question.

L343 – 345: It remains unclear how they correspond to the symptoms of Crohn's disease.

We apologize for not being sufficiently clear at this point. We now clearly refer to Table S3 last column which lists the corresponding data for human symptoms of Crohn’s disease.

Page 10, line 286-287: *The sensitivity and specificity of all findings were calculated and compared to corresponding data for human Crohn’s disease (Tab. S3, last column).*

L387 – 388 (Fig. 9F-I): Why are figure 9F-I not involved in Figure 7?

We have thought about that but decided to include 9F-I into Figure 9, since this Figures deals with the microbiome of *Manduca sexta*, and in this context Figures F-I demonstrate how the microbiome is altered in the presence of HOCl.

L424 – 428 “Overall, these findings demonstrate that our approach is,, testing and high-throughput screening of new therapeutic concepts.”: But, as the authors are also discussing, it remains under investigation whether DUOX is involved in IBD.

Yes, we agree that it remains under investigation whether DUOX is involved in IBD. Nevertheless, we have demonstrated in our study that preclinical hypothesis testing is within the scope of our platform (as reasonably indicated with *M. sexta* DUOX). Clearly, further (pre)clinical studies are required to support this notion: The results from our screening system should inspire people working with mice to more focused studies and to create specific mouse lines to further confirm the hypothesis derived from insect studies.

L527 – 543 A possible role of DUOX in IBD: Although this prediction is interesting, the hypothesis should be confirmed by experiments.

We agree that this question should be further addressed and we have tried to promote this preclinical hypothesis with as much data as possible. However, we feel that further confirmation in a murine or clinical setting is beyond the scope of our work.

References for Reviewer #1

- 1 Carter, M. J., Lobo, A. J., Travis, S. P. & Ibd Section, B. S. o. G. Guidelines for the management of inflammatory bowel disease in adults. *Gut* **53 Suppl 5**, V1-16 (2004).
- 2 Dubois-Camacho, K. *et al.* Glucocorticosteroid therapy in inflammatory bowel diseases: From clinical practice to molecular biology. *World J. Gastroenterol.* **23**, 6628-6638 (2017).
- 3 Sajjadian, S. M. & Kim, Y. PGE2 upregulates gene expression of dual oxidase in a lepidopteran insect midgut via cAMP signalling pathway. *Open Biology* **10**, 200197 (2020).
- 4 Dean, P. *et al.* Modulation by eicosanoid biosynthesis inhibitors of immune responses by the insect *Manduca sexta* to the pathogenic fungus *Metarhizium anisopliae*. *J. Invertebr. Pathol.* **79**, 93-101 (2002).
- 5 Mandato, C. A., Diehl-Jones, W. L., Moore, S. J. & Downer, R. G. The effects of eicosanoid biosynthesis inhibitors on prophenoloxidase activation, phagocytosis and cell spreading in *Galleria mellonella*. *J. Insect Physiol.* **43**, 1-8 (1997).
- 6 Sajjadian, S. M. & Kim, Y. Dual Oxidase-Derived Reactive Oxygen Species Against *Bacillus thuringiensis* and Its Suppression by Eicosanoid Biosynthesis Inhibitors. *Front. Microbiol.* **11**, 528 (2020).
- 7 Broderick, N. A., Raffa, K. F. & Handelsman, J. Chemical modulators of the innate immune response alter gypsy moth larval susceptibility to *Bacillus thuringiensis*. *BMC Microbiol.* **10**, 129 (2010).
- 8 Kumar, S., Molina-Cruz, A., Gupta, L., Rodrigues, J. & Barillas-Mury, C. A peroxidase/dual oxidase system modulates midgut epithelial immunity in *Anopheles gambiae*. *Science* **327**, 1644-1648 (2010).
- 9 Jang, S. *et al.* Dual oxidase enables insect gut symbiosis by mediating respiratory network formation. *Proc. Natl. Acad. Sci. U. S. A.* **118** (2021).

Response to Reviewer #2

Overall Assessment:

In this manuscript, the authors applied diagnostic imaging technologies to hornworm larvae as tools to investigate intestinal pathologies caused by infection, commensal dysbiosis, and colitis-inducing agents. They propose to use these imaging technologies of insect larvae as a high-throughput screening platform for the identification of new effectors and inhibitors of gut inflammation, new antimicrobials, and study of host-pathogen interactions among other applications. The authors used 3 cases to illustrate the functionality of their approach: Bt infection, uracil treatment, and infection with a native member of hornworm flora. Overall, this is a well-performed study with an enormous amount of data. It establishes a new high-throughput screening platform in insects that can be used for a variety of purposes and is of potential interest to researchers in several fields. I have a few suggestions that can help to improve the manuscript.

Thank you for this positive feedback!

Main comments:

1. This is a very data-rich manuscript, however many of the results are not discussed at all or just mentioned very lightly. The authors should make an effort to better explain their results and make sure that every panel of each figure is mentioned in the text (this is not the case now).

Thank you for making us aware of this shortcoming. We have carefully revised the entire manuscript to make sure that now every panel is mentioned.

For example, I would appreciate an explanation how FDG is used to evaluate metabolism (line 236).

We agree and now clearly refer at this point to **Fig. S1B** and **Supplementary Video 2** which explain the concept how FDG is used to evaluate metabolism.

Page 6, line 174-175: *In contrast, FDG-PET uses the uptake of 18F-deoxyglucose (FDG) to evaluate tissue metabolism (Fig. S1B; Supplementary Video 2).*

Furthermore, we modified the part in the Discussion section dealing with this issue as follows:

Page 10, line 272-280: *Typically, inflamed tissue takes up more glucose due to the high glycolytic turnover of infiltrated immune cells. However, we observed the opposite: FDG-PET scans revealed a significantly lower PUVmaxn in the apical region of animals fed with Bt or DSS compared to the control and E. coli groups (Fig. 4C1). Because the midgut and fat body could not be distinguished, we dissected these organs from Bt-infected and control animals and checked them for tissue-specific inflammation-dependent differential FDG uptake, confirming that the PUVmaxn of these organs in Bt-infected animals was lower than the control value (Fig. S16). The severe Bt phenotype with incipient necrosis (Fig. 3D and S14) probably leads to an overall depression of FDG uptake¹ and masks the inflammation-associated effects.*

Also, lines 351-352, this sentence needs a better explanation.

We agree and have formulated the sentence much clearer, now explicitly mentioning necrosis as the reason for the FDG dropdown.

Page 11, line 306-308: *At the metabolic level, FDG uptake decreased further over time, confirming progressive tissue impairment due to necrosis (Fig. 5F).*

2. The authors should organize the figures in a more consequential order or reorganize the text in a way that the readers do not need to go back and forth with the figures.

Here, we kindly disagree: This manuscript comprises 10 main figures, 26 supplementary figures, 17 supplementary tables, and 9 supplementary videos. We have spent an extensive amount of time to organize the figures and findings in a way that is most appealing and comprehensible to the reader. We are aware that it is sometimes required to switch between the various items, but we feel the way we present the data is the most convenient compromise for this.

3. In line 397, the authors refer to DPI as bona fide Nox inhibitor but used it to inhibit Duox. Is there any evidence that DPI inhibits Duox? How do the authors know that the effect that they observed with DPI is due to Duox and Nox inhibition? It would be nice to provide an experimental confirmation that Duox is inhibited by DPI in *M. sexta*.

The Reviewer is correct and this point is crucial. In the literature, there is compelling evidence that DPI is a very potent inhibitor of DUOX²⁻⁴. In fact, it is the reference inhibitor, inhibiting NOX, including DUOX¹². We also confirmed that *M. sexta* DUOX is inhibited by DPI using the R19s assay in **Fig. 6D**⁵⁻⁸. In this context, measuring the R19s signal is a standard method to detect DUOX-dependent HOCl production⁵⁻¹³. After DPI treatment (uracil+DPI+R19s), we found a significantly lower R19s signal than uracil alone (uracil+R19s).

Fig. 6: Conservation of human and *Manduca sexta* DUOX and HOCl production. (D) DUOX-dependent HOCl production after uracil treatment and its inhibition with diphenyleneiodonium (DPI) but not with N-acetylcysteine (NAC).

4. The authors should provide more details on the data behind graphs. Since they keep referring to high-throughput but never mention how many animals were analysed it is not clear how high-throughput it is. What do bar graphs show, e.g. mean+SD? What does each dot mean? How many animals were used? How many repeats? For survival graphs, what do they show: representative experiment? average of n experiments? How many animals were used per treatment?

We apologize for being not sufficiently clear and carefully revised the entire manuscript to address these concerns of this reviewer. Every figure now contains a statement addressing the means, the SD and data points. In addition, survival graphs now list the number of animals used, and in the Material and Methods sections we clarify that the survival kinetics show the sum of the conducted experiments. A total of 808 animals were examined and **Fig. 10** illustrates the individual numbers for the respective experiments.

Page 28, 33 and 37; line 599-600, 689-690 and 794-795: Survival kinetics show the sum of the conducted experiments. Bar charts represent mean and SD. Every data point represents a single animal.

Supplement, page 56, line 226-229: *Survival kinetics were subject to Kaplan-Meier survival analysis with a log-rank (Mantel-Cox) test to detect differential survival. Each survival experiment was done in triplicate. The Kaplan-Meier plots show the sum of these experiments. The n refers to the number of animals used for the respective analysis.*

Fig. 10: Experimental procedure of CT, MR, and PET imaging used in this study

5. Western blot in Fig. 7B, 7C. Was a specific gut region taken for quantification or all regions were combined? Explain in the legend. Since there is no loading control, how the authors can be sure that the differences are not due to differences in the loaded amount?

We apologize for any confusion and have revised the figure as well as our analysis to present the results in a more convenient way. In order to streamline this figure, we removed the data after 34 h uracil exposure since they did not improve the understanding of the overall results.

To confirm the Western blot results, we performed a quantitative real-time RT-PCR analysis after 8 h of uracil/control exposure with the same sample regime as used in the original experiments (FG: foregut, aMD: anterior midgut, MD: medial midgut, pMD: posterior midgut, and II: ileum, part of the hindgut; see also below) employing the *MsDUOX* primers given in **Tab. S1**. The qPCR results are now incorporated in **Fig. 7D** and **S23A+B** and revealed an induction of *MsDUOX* in the anterior digestive tract, which may be due to the fact that this part of the gut is first exposed to the bolus with uracil. These results are in line with our Western blot analysis, which showed a strong DUOX signal in the posterior midgut of most animals and upregulation in the anterior midgut of animals exposed to uracil (**Fig. 7B** and **Fig. S22 E+F**).

Furthermore, we provide now a more transparent evaluation of the Western blots based on the presence or absence of a DUOX signal per animal (**Fig. 7C** and **Fig. S22**). Similar as above, foregut (FG), the anterior midgut (aMD), the medial midgut (MD), the posterior midgut (pMD), and the hindgut (HG) were sampled for each animal. In the obtained Western blots, we counted the presence or absence of the DUOX signal and reported the absolute number of DUOX-positive gut samples per animal. Coomassie staining confirmed similar loading of gels from control/uracil samples, while Ponceau S staining demonstrated successful sample transfer to the membrane, as shown by the uniformly occurring band at 30 kDa (**Fig. S22 H-K**, dashed

white rectangles). All gut samples (FG, aMD, MD, pMD, and HG) from the uracil-exposed animals showed a DUOX signal (15 out of 15), but only 13 samples from control animals (out of 20 samples) showed a DUOX signal. Based on these data, we performed a Mann-Whitney U test and reported a significant upregulation of DUOX after uracil treatment, which is also known from other arthropods¹³⁻²⁰.

Supplement, page 61-62, line 356-371: **Quantitative real-time RT-PCR analysis:** Total RNA was prepared from different gut regions (foregut, anterior-, median-, posterior-midgut and ileum) of 6 control or 6 animals exposed to uracil (8 h) and from pools of tissues (head, fat, central nervous system, labial gland, muscle, skin, trachea and Malpighian tubules) from *M. sexta* fifth instar larvae (L5d5) using the RNeasy Mini Kit (Qiagen) followed by DNase I digestion (Thermo Scientific, Waltham, USA). cDNA synthesis was performed using the biotechrabbit cDNA Synthesis Kit and oligo(dT) primers following the manufacturers' instructions (biotechrabbit GmbH, Berlin, Germany). The qPCRs were performed in triplicates with the qTOWER3 (analytikjena, Jena, Germany) using qPCR SyGreen Mix Fluorescein (Nippon Genetics, Tokyo, Japan), 100 ng of the respective cDNA and pairs of *MsDUOX* specific primers (forward primer: 5'-AAGCACTTCGAGTGGTTCATC-3'; reverse primer: 5'-TCAAGAAGGAGGACATGTCG-3', Table S1). Relative gene expression was calculated based on the comparison of CT values for *MsDUOX* and the reference gene *MsEF1a* (forward primer: 5'-CTTCACAGCTCAGGTCATCG-3'; reverse primer: 5'-GAAGGACTCCACACACATGG-3', Table S1). The specificity of the PCR was confirmed by melting-curve analysis and mean normalized expression was determined according to²¹. Mean-normalized *MsDUOX* expression values of the anterior digestive tract (FG, Foregut; aMD, anterior midgut; and MD, medial midgut) or the posterior digestive tract (pMD, posterior midgut and Il, ileum, part of the hindgut) were reported.

Table S1: Sequences of RT-PCR primers (5' to 3' direction).

Gene	GenBank accession number	Primer	Sequence	Product size (bp)
MsEF1 (α)	AF234571.1	Forward	5'-CTTCACAGCTCAGGTCATCG-3'	229bp
		Reverse	5'-GAAGGACTCCACACACATGG-3'	
MsAttacin 1	DQ072728.1	Forward	5'-CCTGTCGTGCCTCTTCCTC-3'	751bp
		Reverse	5'-GAGCGAGGTGGTCTTGTC-3'	
MsGloverin	AM293324.1	Forward	5'-GAAGGTCTTCGGAACCTCTGG-3'	352 bp
		Reverse	5'-CTGGAAGAGACCTTGAAGC-3'	
MsDUOX	MK983103.1	Forward	5'-AAGCACTTCGAGTGGTTCATC-3'	228 bp
		Reverse	5'-TCAAGAAGGAGGACATGTCG-3'	

Supplement, page 62-63, line 389-394: [...] the DUOX signal was assessed as present or absent. Then, the membranes were hydrated in 100% methanol, washed with TBS+T, and rinsed with ultrapure water. Next, the membranes were stained with Ponceau S staining solution (Cell Signaling, Danvers, MA) and a protein with a rel. molecular weight of 30 kDa was used as a loading control. Finally, the number of DUOX-positive gut samples of control and uracil-exposed animals were compared (Fig. 7B+C and S22).

Fig. 7: Location of *M. sexta* DUOX in the gut, its upregulation and effect

(A) *M. sexta* DUOX is localized in the brush border of the midgut. (B) Western blots confirming the upregulation of *M. sexta* DUOX after uracil treatment (C). FG: foregut, aMD: anterior midgut, MD: medial midgut, pMD: posterior midgut, and HG: hindgut. * indicates a protein with the rel. molecular weight of 170 kDa (DUOX). More details about the Western blot analysis are given in Fig. S22. (D) Quantitative RT-PCR analysis of *MsDUOX* demonstrating the induction of *M. sexta* DUOX after uracil treatment (8 h) in the anterior digestive tract (FG, aMD, and MD). More details about the qPCR analysis are given in Fig. S23.

Fig. S22: Western blot with the anti-*M. sexta*-DUOX antibody of different gut regions

(A–C) Illustrations of the presence of *M. sexta* DUOX based on the Western blots E–F. (D) The anatomy of *M. sexta* is given as a reference. (E–F) Western blot analysis with anti-DUOX LLR of the digestive systems of *M. sexta* 24 h after uracil treatment. FG: foregut, aMD: anterior midgut, MD: medial midgut, pMD: posterior midgut. The hindgut (HG) was used as a control (always right lanes). * indicates a protein with the rel. molecular weight of 170 kDa (DUOX). (G) The number of DUOX-positive samples was significantly higher in animals exposed to uracil (control: n = 4, uracil: n = 3). (H–I) Coomassie staining confirmed similar loading of gels from control/uracil samples, while (J–K) Ponceau S staining demonstrated successful sample transfer to the membrane, as shown by the uniformly occurring band at 30 kDa (dashed white rectangles). The # identifies every animal in this experiment and allows comparison of SDS-PAGE and Western blot results. Error bars represent standard deviation.

Fig. S23: Quantitative RT-PCR analysis of *MsDUOX* in the gut and different tissues

(A) Mean-normalized *MsDUOX* expression values of the anterior digestive tract (FG, Foregut; aMD, anterior midgut; and MD, medial midgut) or the posterior digestive tract (pMD, posterior midgut and Il, ileum, part of the hindgut) (C). (D) Mean-normalized *DUOX* expression values of different tissues.

Table S1: Sequences of RT-PCR primers (5' to 3' direction).

Gene	GenBank accession number	Primer	Sequence	Product size (bp)
MsEF1 (α)	AF234571.1	Forward	5'-CTTCACAGCTCAGGTCATCG-3'	229bp
		Reverse	5'-GAAGGACTCCACACACATGG-3'	
MsAttacin 1	DQ072728.1	Forward	5'-CCTGTCGTGCCTCTTCCTC-3'	751bp
		Reverse	5'-GAGCGAGGTGGTCTTGTC-3'	
MsGloverin	AM293324.1	Forward	5'-GAAGGTCTTCGGAACCTG-3'	352 bp
		Reverse	5'-CTGGAAGAGACCTTGAAGC-3'	
MsDUOX	MK983103.1	Forward	5'-AAGCACTTCGAGTGGTTCATC-3'	228 bp
		Reverse	5'-TCAAGAAGGAGGACATGTGC-3'	

Specific comments:

6. Fig. 4H lacks statistical analysis.

Sorry for missing this. We have now included the statistics for **Figure 4H** (which is now **Figure 4G**)

Fig. 4G: High-throughput imaging of larvae exposed to different challenges and concentration-dependent rescue of the colitis-like phenotype with gentamicin

Contrast-enhanced CT and MRI gut wall thickness showed a gentamicin concentration-dependent reduction of gut wall thickness in animals fed with Bt and two different gentamicin concentrations (F and G). This finding was also confirmed by the differential survival of the treated animals (H).

Line 195. Toll is also an NF- κ B pathway, did the authors mean Relish?

Thank you for making us aware of this ambiguity – you are correct: It is imd, not NF- κ B²²⁻²⁵. We have rephrased this sentence accordingly and it now reads:

Page 4, line 133-134: *This includes the insect imd and Toll pathway, which resemble the mammalian TNF- α and TLR pathways*^{25,26}.

References for Reviewer #2

- 1 Rakheja, R. *et al.* Necrosis on FDG PET/CT correlates with prognosis and mortality in sarcomas. *Am. J. Roentgenol.* **201**, 170-177 (2013).
- 2 Augsburger, F. *et al.* Pharmacological characterization of the seven human NOX isoforms and their inhibitors. *Redox Biol.* **26**, 101272 (2019).
- 3 Ha, E.-M., Oh, C.-T., Bae, Y. S. & Lee, W.-J. A direct role for dual oxidase in *Drosophila* gut immunity. *Science* **310**, 847-850 (2005).
- 4 Oliveira, J. H. *et al.* Blood meal-derived heme decreases ROS levels in the midgut of *Aedes aegypti* and allows proliferation of intestinal microbiota. *PLoS Pathog.* **7** (2011).
- 5 Chen, X. *et al.* Synthesis of a highly HOCl-selective fluorescent probe and its use for imaging HOCl in cells and organisms. *Nat. Protoc.* **11**, 1219-1228 (2016).
- 6 Zhang, Y. R. *et al.* A ratiometric fluorescent probe for sensing HOCl based on a coumarin-rhodamine dyad. *Chem. Commun. (Camb.)* **50**, 14241-14244 (2014).
- 7 Chen, X. *et al.* A specific and sensitive method for detection of hypochlorous acid for the imaging of microbe-induced HOCl production. *Chem. Commun. (Camb.)* **47**, 4373-4375 (2011).
- 8 Hachfi, S., Benguettat, O. & Gallet, A. Hypochlorous Acid Staining with R19-S in the *Drosophila* Intestine upon Ingestion of Opportunistic Bacteria. *Bio-protocol* **9**, 1-10 (2019).
- 9 Lee, K.-A., Kim, B., You, H. & Lee, W.-J. Uracil-induced signaling pathways for DUOX-dependent gut immunity. *Fly* **9**, 115-120 (2015).
- 10 Lee, K. A. *et al.* Bacterial uracil modulates *Drosophila* DUOX-dependent gut immunity via Hedgehog-induced signaling endosomes. *Cell Host Microbe* **17**, 191-204 (2015).

- 11 Lee, K.-A. *et al.* Inflammation-modulated metabolic reprogramming is required for DUOX-dependent gut immunity in *Drosophila*. *Cell Host Microbe* **23**, 338-352. e335 (2018).
- 12 Kim, E.-K. *et al.* Bacterial nucleoside catabolism controls quorum sensing and commensal-to-pathogen transition in the *Drosophila* gut. *Cell Host Microbe* **27**, 345-357. e346 (2020).
- 13 Lee, K.-A. *et al.* Bacterial-derived uracil as a modulator of mucosal immunity and gut-microbe homeostasis in *Drosophila*. *Cell* **153**, 797-811 (2013).
- 14 Huang, Y. *et al.* Dual oxidase Duox and Toll-like receptor 3 TLR3 in the Toll pathway suppress zoonotic pathogens through regulating the intestinal bacterial community homeostasis in *Hermetia illucens* L. *PLoS One* **15**, e0225873 (2020).
- 15 Sajjadian, S. M. & Kim, Y. PGE2 upregulates gene expression of dual oxidase in a lepidopteran insect midgut via cAMP signalling pathway. *Open biology* **10**, 200197 (2020).
- 16 Zhang, L., Wang, Y.-w. & Lu, Z.-q. Midgut immune responses induced by bacterial infection in the silkworm, *Bombyx mori*. *Journal of Zhejiang University-SCIENCE B* **16**, 875-882 (2015).
- 17 Yao, Z. *et al.* The dual oxidase gene BdDuox regulates the intestinal bacterial community homeostasis of *Bactrocera dorsalis*. *The ISME journal* **10**, 1037-1050 (2016).
- 18 Chen, Y. *et al.* Molecular characterization of the dual oxidase (LvDuox) gene from the pacific white shrimp *Litopenaeus vannamei*. *Invertebrate Survival Journal* **15**, 316-326 (2018).
- 19 Sun, Z. *et al.* Dual oxidases participate in the regulation of hemolymph microbiota homeostasis in mud crab *Scylla paramamosain*. *Dev. Comp. Immunol.* **89**, 111-121 (2018).
- 20 Ma, Z., Wang, Y., Li, C., Pan, G. & Zhou, Z. Sequence characteristics, expression pattern and pathogen-induced expression of BmDUOX from the silkworm, *Bombyx mori* (Lepidoptera: Bombycidae). *Acta Entomol. Sin.* **60**, 1255-1265 (2017).
- 21 Simon, P. Q-Gene: processing quantitative real-time RT-PCR data. *Bioinformatics* **19**, 1439-1440 (2003).
- 22 Buchon, N., Silverman, N. & Cherry, S. Immunity in *Drosophila melanogaster*--from microbial recognition to whole-organism physiology. *Nat. Rev. Immunol.* **14**, 796-810 (2014).
- 23 Valanne, S., Wang, J. H. & Ramet, M. The *Drosophila* Toll signaling pathway. *J. Immunol.* **186**, 649-656 (2011).
- 24 Myllymaki, H., Valanne, S. & Ramet, M. The *Drosophila* imd signaling pathway. *J. Immunol.* **192**, 3455-3462 (2014).
- 25 Galenza, A. & Foley, E. Immunometabolism: insights from the *Drosophila* model. *Dev. Comp. Immunol.* **94**, 22-34 (2019).
- 26 Zhang, R. *et al.* Toll9 from *Bombyx mori* functions as a pattern recognition receptor that shares features with Toll-like receptor 4 from mammals. *Proc. Natl. Acad. Sci.* **118** (2021).

Response to Reviewer #3

Overall Assessment:

The manuscript by Windfelder et al. provides an in-depth analysis regarding the suitability of using caterpillars as a model for gut inflammation that might be relevant to human health. The authors show that preclinical imaging modalities, particularly CT, can serve as a rapid diagnostic tool for gut health in an insect system, which may serve as a suitable alternative to traditionally used mammalian models. They go on to validate their model and to show further similarity at the molecular level between insect and mammalian (particularly humans) proteins known to be important for modulating gut homeostasis (DUOX) and the response to bacterial pathogens. Overall, the amount of data included in this paper is impressive, and while this reviewer would have appreciated perhaps a more streamlined approach to the resulting figures and supplementary material, I do appreciate the quality of the work and the tone of the conclusions that are drawn from them. While some mammalian folks may not recognize the significance of these results, I do believe that the authors have made a good case for using *M. sexta* as a model system, especially if the major concerns that I have outlined below can be addressed.

Thank you – we were very happy to read this positive assessment! We appreciate your kind and constructive suggestions, that helped us to improve the manuscript.

Main comments:

1. What is the actual spatial resolution of the CT images? The voxel sizes reported here (350 μm for CT, 7.55 μm for u-CT) describe the properties of the instrument (detector size, geometry) and do not accurately reflect the actual spatial resolution of the images. Actual resolution in CT & u-CT can be influenced by many different factors, and may be slightly different from one scan to the next even with identical imaging parameters. I would advise the authors to use a phantom designed to determine actual spatial resolution at the imaging conditions used. The phantoms provided in Fig. S4 do not satisfy this criteria, as they only show that the resolution is sufficient to ~ 4 mm.

We agree and performed a number of new measurements to meet this concern of the reviewer. To this end, we designed and printed a duplicate of the inner part of the micro-PET hot rod 3D Derenzo phantom with six different sets of bores using a Anycubic Photon Mono X 3D printer with phrozen ABS-like creamy white monomers. All dimensions of the printed phantom (diameter, height and 6 holes 0.6, 0.8, 1.0, 1.2, 1.5, 2.0 mm) were according to the PTW manufacturer specifications. We have chosen the 0.2 mm difference in bore size because the smallest difference in mean contrast-enhanced CT gut wall thickness found was 0.26 mm (0.1 M uracil vs. control, **Tab. S10**). The bore size of 1 mm, and the bore size of 1.2 mm are reflecting an even smaller difference than the smallest mean differences in this study. Subsequently, we measured the bores in the Derenzo phantom with the two scanners utilized in this study (Siemens SOMATOM Force and Siemens SOMATOM Emotion 6) using the same settings applied to image the caterpillars (**Reviewer Fig. I**). Further, we used the suggested FWHM thickness measurements to determine the bore size. **Reviewer Fig. I** shows that all bores could be differentiated, including bores with a size of 1 mm and 1.2 mm. The double mean of all bore measurements was 0.017 mm using the SOMATOM Force and 0.034 mm using the SOMATOM Emotion 6, suggesting the capability to measure multivoxel objects with 200 μm size difference well below the voxel size with FWHM thickness measurements.

In addition, we have used a group of different CT scanners available to us and measured the Derenzo phantom with the μCT Skyscan 1173, the photon-counting CT NAEOTOM Alpha,

the dual source CT SOMATOM Force, the signal source CT SOMATOM X.ceed, the signal source CT SOMATOM go.Top and the signal source CT SOMATOM Emotion 6. All bores in the Derenzo phantom could be differentiated on all scanners, confirming the capability of commonly available clinical CT scanners to detect small differences in gut wall thickness. These data are now included into the revised manuscript as **Fig. S6**.

Reviewer Fig. I: 3D-printed Derenzo phantom in two different clinical CT scanners and the corresponding FWHM-thickness measurements of the bores using the same settings used to image the caterpillars. (A) the dual source CT SOMATOM Force and **(B)** the signal source CT SOMATOM Emotion 6.

Fig. S6: 3D-printed Derenzo phantom in a μ CT or five different clinical CT scanners and the corresponding FWHM thickness measurements of the bores. (A-A1) Derenzo phantom in the μ CT Skyscan 1173, (B-B1) the photon-counting CT NAEOTOM Alpha, (C-C1) the dual source CT SOMATOM Force (D-D1), the signal source CT SOMATOM X.ceed, (E-E1) the signal source CT SOMATOM go.Top and (F-F1) the signal source CT SOMATOM Emotion 6.

Finally, we reviewed our previous phantom measurements using glass capillaries (which already had shown that thickness differences of 1 mm and 1.2 mm could be detected with the imaging settings used). Here, in analogy to the gut wall thickness measurements, we measured the thickness of the 1 mm thick glass capillary using the FWHM measurements suggested. The

mean deviation of our measurements (n=20) was 52 μm , and the double mean of the SD 0.07 mm (Fig. S5).

Fig. S5: Validation of CT resolution using a capillary glass phantom.

(A) CT images of the capillary glass phantom. (B) FWHM measurement of the capillary wall. Error bars represent standard deviation.

2. The actual spatial resolution is important because it can directly impact the accuracy of the thickness measurements. For example, significant differences in gut thickness is reported (in Fig. 1M, 4A1, 4D, 4F, for example) that vary by ~250-400 microns. Assuming that the actual spatial resolution of the CT scans are ~ 1 millimeter, this puts the actual size of the gut walls themselves either at or below the resolution limit. At the minimum, this could directly impact the ability to detect less severe (but nonetheless biomedically relevant) forms of gut inflammation and should be addressed accordingly in the manuscript.

We fully agree with this statement and it is important to address this point. However, we kindly disagree with the assumption of a CT resolution of ~1 mm. All CTs used and tested have a resolution well above this value ($\text{MTV}(0\%) > 17 \text{ lp/cm}$), which is smaller than the voxel size of 350 μm . While our 3D phantom measurements do not allow to quantify the CT images' actual resolution, they demonstrate that edge detection with FWHM measurements of cylindrical objects is possible to a level of differences $d \sim 0.2 \text{ mm}$ (see above). Less severe inflammation maybe nonetheless biomedically relevant resulting in gut thickness differences below 0.2 mm. Such small thickness differences are more challenging to detect. Here, further optimizations such as reducing the voxel size are necessary. But usually, increasing the resolution comes with the price of a smaller FOV. In this context, the newest generation of photon counting clinical CT scanners (e.g., the NAEOTOM Alpha) could be helpful¹⁻⁴. Also, the emergence of CT super resolution is of particular interest and could help to resolve even slight gut wall thickness differences in large cohorts of animals⁵. In the revised version of our manuscript, we have addressed these considerations accordingly:

Page 16, line 431-440: *These measurements were validated via semi-automatic FWHM thickness measurements, and their reliability was confirmed by test-retest analysis (Fig. S7–S9). Of note, the disease models used in this study induced moderate to severe alterations in gut wall thickness as compared to control animals which could easily be resolved by our current approach. However, if the induced effects are weaker, optimization of the resolution may be required. For this, the larvae's long and straight body shape allow densely packed experimental setups with small FOVs (Fig. 1). Furthermore, instead of averaging all CT and MRI parameters over the entire midgut (as in the present study), a regional or sub-regional analysis might be advantageous if the colitis-like phenotype is restricted to certain midgut areas.*

3. It is also unclear to me how the thickness measurements were made for CT and MRI. The Methods state “The maximum gut wall thickness measured in contrast-enhanced T1-weighted MRI was measured at the thickest visible spot on each slice”; with the CT measurements “as described for MRI”. Were the measurements done by manually drawing a line across the gut? How were boundaries for these determined—was a certain

pixel intensity threshold used? This could be highly error prone, especially if the resolution of the tomograms are not sufficiently high (a single pixel could therefore throw off a measurement by 100 microns or more). Please clarify this in the Methods section.

Since this and the next two points are tightly related, we address these comments together.

Yes, the gut wall thickness measurements were done manually. First, we measured the maximum gut wall thickness on each slice, which resulted in >80 measurements per animal, and averaged all values per midgut. The averaged measurements are robust against measurement errors and highly comparable among animals, but they are also labor-intensive and do not allow for regional thickness differences. With regard to comment #4, we followed the suggestion of this reviewer to validate the manual thickness measurements via FWHM measurements and incorporated these data in **Fig. S7+S8** (Figures see next page).

In CT and MRI, FWHM and manual measurements were highly correlated and comparable to the previous measures (**Fig. S7A+B** and **S8A+B**). We also performed a Bland Altman analysis comparing manual CT gut wall thickness and FWHM measurements. The measurements are within the 95% limit of agreement and can be used interchangeably (**Fig. S7C** and **S8C**).

In addition, as suggested, we significantly increased the number of animals included in our μ CT analysis. Now the study includes measurements of complete midguts from 10 animals with a resolution of 7.5 μ m. Both the maximum PTA-stained gut wall thickness from the μ CT and the macroscopic maximum gut wall thickness measurements from the clinical CT scanner are in good agreement (**Fig. 2C+D**).

Taken together, we added multiple experiments that validated our thickness and thickness difference measurements: We feel confident that with this we adequately addressed and overcame the concerns of this reviewer about spatial resolution.

Page 41, line 894-897: *The maximum gut wall thickness measured in contrast-enhanced T1-weighted MRI was measured manually at the thickest visible spot on each slice. The accuracy of these measurements was verified through a Full Width at Half Maximum (FWHM) analysis (Fig. S7+S8).*

Page 16, line 431-433: *These measurements were validated via semi-automatic FWHM thickness measurements, and their reliability was determined by test-retest testing (Fig. S7-S9).*

Fig. S7

Fig. S7: Comparison of manual CT gut wall thickness and FWHM measurements

(A) Comparison of manual CT gut wall thickness and FWHM measurements in control and Bt animals. (B) Correlation of FWHM and manual measurements. (C) CT images with FWHM measurements. (D) Bland Altman plot comparing manual CT gut wall thickness and FWHM measurements. Bar charts represent mean and SD. Every data point represents a single animal.

Fig. S8

Fig. S8: Comparison of manual MRI gut wall thickness and FWHM measurements

(A) Comparison of manual MRI gut wall thickness and FWHM measurements in control and Bt animals. (B) Correlation of FWHM and manual measurements. (C) MR images with FWHM measurements. (D) Bland Altman plot comparing manual MR gut wall thickness and FWHM measurements. Bar charts represent mean and SD. Every data point represents a single animal.

Fig. 2C and D: (C) PTA-stained midgut in μCT. (D) Mean CT (n = 25, voxel size of 350 μm) and μCT (n=10, voxel size of 7.55 μm) gut wall thicknesses from the complete midgut with descriptive statistics.

4. Considering the issues outlined above, it might be preferable to perform a Full Width Half Maximum (FWHM) analysis of line scans across the gut wall for the width measurements in order to verify the accuracy of these measurements, and also lessen the concern about the resolution limits imposed by CT and the size of the gut wall. FWHM measurements do not rely on manual determination of signal boundaries and instead rely on mathematical Gaussian fits to the line scan data. Also, FWHM measurements have become the standard approach for measuring optically unresolved structures in fluorescent light microscopy (standard vs. super resolution for example) to provide accurate and unbiased parameters for resolution comparisons below the diffraction limit. Please see above our answer to your main comment #3.

Specific comment:

Pg 8, lines 3-5: The sample size for micro-CT measurements are low. A higher n would give further confidence in the CT image measurements despite the potential spatial resolution issue outlined above.

Please see above our answer to your main comment #3.

Pg 6., lines 1-14: I think this argument could be further strengthened by incorporating a statement on the availability/use of genetic tools in *M. sexta* to help elucidate molecular mechanisms.

Thank you, that is an excellent point. We now mention Fandino *et al.* 2019 and Elefterianos *et al.* 2006, who used CRISPR-Cas9 and RNAi to explore molecular mechanisms in *M. sexta*^{6,7}.

Page 5, line 153-154: *The availability of genetic tools in M. sexta helps to elucidate pathological processes down to a molecular level*^{6,7}.

Would the inability to detect a significant increase in CT signal density for DSS treatment (Fig. 4A2) despite a significant increase in gut wall thickness limit the applicability of CT for mild models of gut inflammation?

According to our ROC curve analysis (**Tab. S3**, see next page), CT signal attenuation has a lower sensitivity (84.62%) and specificity (58.33%) compared to CT gut wall thickness measurements (sensitivity of 96.00% and specificity of 88.89%). Therefore, mild models of gut inflammation should be analyzed with modalities exhibiting with high sensitivities for gut wall thickness measurements (like CT or MRI). Clearly, the availability of additional imaging modalities is advantageous and helps to validate the results.

Table S3: Diagnostic findings after Bt, DSS 5% and *E. coli* treatment with corresponding threshold values.

Diagnostic finding CT, MR or PET	Change after Bt treatment (compared to control)			Change after DSS 5% treatment (compared to control)			Change after E. coli treatment (compared to control)		Diagnostic values				
	without threshold (1-way-ANOVA or Kruskal-Wallis ANOVA)	above or under threshold? (Chi-squared test)	or	without threshold (1-way-ANOVA or Kruskal-Wallis ANOVA)	above or under threshold? (Chi-squared test)	or	without threshold (1-way-ANOVA or Kruskal-Wallis ANOVA)	above or under threshold? (Chi-squared test)	Threshold value	Sensitivity	Specificity	ROC Area	ROC Area from the Literature of human Crohn's disease
CT gut wall thickness (post-iodixanol)	↑**** <0.0001	↑**** <0.0001		↑*** 0.0005	↑** 0.0022		ns	ns	>1.006 mm	96.00%	88.89%	0.97	CT: 0.90 - 0.89 Lee et al. 2009 ³⁰
CT signal density (post-iodixanol)	↑* 0.0132	↑* 0.0254		ns	↑** 0.0012		ns	↑** 0.0076	> 84.23 Hu	84.62%	58.33%	0.80	
MR gut wall thickness (post-gadolinium)	↑**** <0.0001	↑**** <0.0001		↑**** <0.0001	↑**** <0.0001		ns	ns	>1.833 mm	80.77%	85.37%	0.86	MRI: 0.93 - 0.95 Lee et al. 2009 ³⁰
MR normalized T1 signal (post-gadolinium)	↑**** <0.0001	↑**** <0.0001		ns	ns		ns	ns	>1.883	90%	80.47%	0.91	
MR T2 signal	↑* 0.0221	↑** 0.0010		ns	ns	/	/	/	>997.6	85%	68.75%	0.78	
FDG-PET PUV_{max}	↓*** 0.0010	↓**** <0.0001		↓** 0.0054	↓* 0.047		ns	↑* 0.0282	<0.5405 kg/MBq	84%	65.52%	0.72	PET/CT: 0.85 Louis et al. 2007 ³¹

In an effort to streamline the main figures, I would suggest moving Fig. 5 to the supplement, at least 5A-F.

We are well aware of the extensive character of this study. However, Fig. 5A-E and especially Fig. 5F help to understand the validation of the imaging findings. The drop in FDG uptake in the Bt phenotype was, at first glance, surprising and in contradiction to mammalian models of gut inflammation. Here, Fig. 5F helps, together with the scanning electron microscopic analysis (Fig. 3C+D), to understand the gradual drop of FDG uptake most likely due to necrosis. Thus, we would prefer to keep Fig. 5 unchanged.

Pg. 19, line 3: later should be latter.

Thank you for this correction.

Pg. 24, lines 9 & 11. Italicize *M. sexta*

Sorry, for the incorrect formatting. Corrected accordingly.

Pg. 25, Figure legend: Remove uracil from title, as this figure is just for BT treatment?

Thank you for making us aware of this. We removed uracil from the title.

Page 25, line 571-572: Fig. 3 Histopathological characterization of the colitis-like phenotype in larvae treated with *Bacillus thuringiensis* (Bt)

Pg. 37. Consider moving Fig. 10 to supplement

We already considered this, but the comments from Reviewer #2 made us aware of some uncertainties regarding the number of animals used in this study. Since Figure 10 provides a good overview of our experiments and the exact number of animals used in each experiment, we would like to leave this figure in a prominent place.

Pg. 39-40. It is not clear to me exactly how the thickness measurements were made? And what exactly were the 3 different curve fitting tests that were used to identify differences in the CT gut wall thickness curves?

Sorry for being not sufficiently clear. We again used the CT gut wall thickness measurements, but this time not the averaged values: Instead, we used the thickness measurement for each CT slice and plotted the mean values per treatment as a dashed line (**Fig. 4D**). Next, we compared the resulting treatment-specific sequential gut wall thickness curves (c=12, u=22, u+DPI=22, u+NAC=23, Bt=13, DSS=14, *E. coli*=12) with an extra sum-of-squares F test and verified whether each treatment had the same slope as well as intercept and could be represented as a single global model (H0) or if individual models for each treatment should be employed (H1). The test yielded an F value of 389.3 (6,7931) and a p value of <0.0001. Based on this, H0 was rejected, and the dataset is best represented with different treatment-specific sequential gut wall thickness curves. Finally, we fitted a horizontal line (the respective mean thickness value of each treatment) to each dataset to make the curves more accessible and comparable.

We updated the Material and Methods section as follows:

Supplement, page 51-52, line 107-114: *A subset of animals without gaps in the gut wall thickness measurements along the whole midgut was selected for sequential treatment-specific analysis of the axial CT scans. For better comparability of the thickness measurements, the first 10 measurements were excluded. The resulting treatment-specific sequential gut wall thickness curves were compared with an extra sum-of-squares F test and tested whether each treatment had the same slope and intercept and could be represented as a single global model (H0) or if individual models for each treatment should be employed (H1).*

Fig. 4D

Fig. 4D. Mean sequential CT gut wall thickness curves (the horizontal lines represent the overall treatment-specific mean thickness).

Fig. 8D

Fig. 8D. Mean sequential CT gut wall thickness curves (the horizontal lines represent the overall treatment-specific mean thickness).

Fig S3 legend: Plains=Planes

Thank you for this correction.

References for Reviewer #3

- 1 Willemink, M. J., Persson, M., Pourmorteza, A., Pelc, N. J. & Fleischmann, D. Photon-counting CT: Technical Principles and Clinical Prospects. *Radiology* **289**, 293-312 (2018).
- 2 Flohr, T. *et al.* Photon-counting CT review. *Phys. Med.* **79**, 126-136 (2020).
- 3 Rajendran, K. *et al.* First Clinical Photon-counting Detector CT System: Technical Evaluation. *Radiology* **303**, 130-138 (2022).
- 4 Thomsen, F. S. L., Horstmeier, S., Niehoff, J. H., Pena, J. A. & Borggrefe, J. Effective Spatial Resolution of Photon Counting CT for Imaging of Trabecular Structures is Superior to Conventional Clinical CT and Similar to High Resolution Peripheral CT. *Invest. Radiol.* **57**, 620-626 (2022).
- 5 Zhang, Z. *et al.* Self-supervised CT super-resolution with hybrid model. *Comput Biol Med* **138**, 104775 (2021).
- 6 Fandino, R. A. *et al.* Mutagenesis of odorant coreceptor Orco fully disrupts foraging but not oviposition behaviors in the hawkmoth *Manduca sexta*. *Proc. Natl. Acad. Sci. U. S. A.* **116**, 15677-15685 (2019).
- 7 Eleftherianos, I., Millichap, P. J., French-Constant, R. H. & Reynolds, S. E. RNAi suppression of recognition protein mediated immune responses in the tobacco hornworm *Manduca sexta* causes increased susceptibility to the insect pathogen *Photorhabdus*. *Dev. Comp. Immunol.* **30**, 1099-1107 (2006).

Response to Reviewer #4

Main comments:

- What are the noteworthy results?

They have tested and validated a model non vertebrate organism for screening innate immune phenotypes by both anatomical and molecular imaging. This fills a gap between fruit flies and nematodes which can be examined by microscopic techniques and rodents that can be molecularly phenotyped by structural and molecular imaging.

Thank you for this positive evaluation!

- Will the work be of significance to the field and related fields? How does it compare to the established literature? If the work is not original, please provide relevant references. Identifying phenotypes that are consistent across size and evolutionary time between this new organism and mice might improve the translational leap between mice and humans. An interesting question to be determined over time.

Thank you.

- Does the work support the conclusions and claims, or is additional evidence needed?

- Are there any flaws in the data analysis, interpretation and conclusions? - Do these prohibit publication or require revision?

While the goal is high throughput screening the analysis of imaging data is primarily focused on groupwise difference comparison. This may be a useful initial step. Analysis of effect sizes, eg z' or at least large cohens d effect sizes are more useful. As would be a deeper dive into test-retest analysis and limits of agreement. Without this deeper dive their central thesis of high throughput screening remains to be validated.

Since this and the next two points are tightly related, we address these comments together.

Thank you for these excellent suggestions. As requested, we carried out the additional analyses. **Tab. S14-S17** document each treatment's effect size (η^2 , **g** or **d**). As can be recognized from these tables, almost all interventions had a large effect, which are definitively favorable for high throughput screenings. Nevertheless, although CT or MR gut wall thicknesses yielded good sensitivity and specificity values, the gut wall thickness has anatomical and physiological constraints, restricting our approach to a certain pathological level. In this context, signal density (in CT) and T1 signal enhancement (in MRI) are not directly limited, making these imaging findings more attractive for high throughput screening. Here, T1 signal enhancement was superior to signal density concerning sensitivity and specificity. Thus, we calculated the Hedges' g for the T1 signal enhancement in our positive control Bt (36h) and compared that with the Hedges' g for MRI gut wall thickness (also for Bt 36h). As expected, the Hedges' g for the T1 signal enhancement ($g = 6.6$) was higher than the Hedges' g for MRI gut wall thickness ($g = 3.5$, **Tab. S14**). Furthermore, we performed a test-retest analysis and compared CT gut wall thickness, signal density, and PUV maxn (**Fig. S9A+C+E**). All three imaging findings showed excellent reliability. Finally, a Bland Altman analysis was carried out with these measurements (**Fig. S9B+D+F**). Also here, all measurements were within the 95% limit of agreement. We have incorporated these important additional statistical information into the revised version of the manuscript:

Page 7, line 197-199: *Furthermore, we validated the thickness measurements of the gut wall with semi-automatic FWHM thickness measurements in CT and MRI (Fig. S7–S8) and verified the test-retest reliability (S9).*

Page 10, line 290-293: *Additionally, we analyzed the effect size of every treatment (Tab. S14–S17) and identified T1 signal enhancement as parameter with the largest value ($g=6.6$, Tab. S14), rendering it as excellent readout for high throughput screenings.*

Table S14: Effect size of the used treatments with the respective imaging modality; eta squared (η^2), Hedges' g or Cohen's d .

Imaging modality	Diagnostic finding	Treatment	Effect size of treatment η^2 , Hedges' g or Cohen's d (compared to control)	Large effect*	Moderate effect*	Small effect*	No effect*
CT	CT gut wall thickness (post-iodixanol)	Bt 12h	$\eta^2 = 0.6010$ $g = 3.38151$	X			
CT	CT gut wall thickness (post-iodixanol)	Bt 36h	$\eta^2 = 0.5059$ $g = 1.985085$	X			
CT	CT gut wall thickness (post-iodixanol)	DSS 5%	$\eta^2 = 0.3801$ $g = 1.526288$	X			
CT	CT gut wall thickness (post-iodixanol)	Uracil 0.1M	$\eta^2 = 0.4951$ $g = 1.93884$	X			
CT	CT gut wall thickness (post-iodixanol)	Uracil 1.0M	$\eta^2 = 0.5273$ $g = 2.022548$	X			
CT	CT gut wall thickness (post-iodixanol)	Microbacterium sp. (#3)	$\eta^2 = 0.8648$ $g = 5.971387$	X			
CT	CT gut wall thickness (post-iodixanol)	Enterococcus sp. (#4)	$\eta^2 = 0.5675$ $g = 2.308153$	X			
CT	CT gut wall thickness (post-iodixanol)	E. coli	$\eta^2 = 0.01557$ $d = 0.244981$				X X
CT	CT signal density (post-iodixanol)	Bt 12h	$\eta^2 = 0.2707$ $d = 1.165844$	X			
CT	CT signal density (post-iodixanol)	Bt 36h	$\eta^2 = 0.8221$ $g = 4.128445$	X			
CT	CT signal density (post-iodixanol)	DSS 5%	$\eta^2 = 0.2188$ $g = 1.017681$	X	X		
CT	CT signal density (post-iodixanol)	Uracil 0.1M	$\eta^2 = 0.4309$ $g = 1.752486$	X			
CT	CT signal density (post-iodixanol)	E. coli	$\eta^2 = 0.1931$ $g = 0.95736$	X			
MRI	MR gut wall thickness (post-gadolinium)	Bt 12h	$\eta^2 = 0.4014$ $g = 1.656781$	X			
MRI	MR gut wall thickness (post-gadolinium)	Bt 36h	$\eta^2 = 0.7520$ $g = 3.521243$	X			
MRI	MR gut wall thickness (post-gadolinium)	DSS 5%	$\eta^2 = 0.3869$ $g = 2.02662$	X			
MRI	MR gut wall thickness (post-gadolinium)	Uracil 0.1M	$\eta^2 = 0.3492$ $g = 1.534149$	X			
MRI	MR gut wall thickness (post-gadolinium)	Uracil 1.0M	$\eta^2 = 0.5273$ $g = 2.022548$	X			
MRI	MR gut wall thickness (post-gadolinium)	E. coli	$\eta^2 = 0.09932$ $g = 0.726905$		X X		
MRI	MR normalized T1 signal	Bt 12h	$\eta^2 = 0.4017$ $g = 1.596851$	X			
MRI	MR normalized T1 signal	Bt 36h	$\eta^2 = 0.9196$ $g = 6.712817$	X			
MRI	MR normalized T1 signal	DSS 5%	$\eta^2 = 0.07492$ $g = 0.588844$		X X		
MRI	MR normalized T1 signal	Uracil 0.1M	$\eta^2 = 0.2248$ $g = 1.04751$	X			
MRI	MR normalized T1 signal	E. coli	$\eta^2 = 0.0006059$ $g = 0.046986$				X X
MRI	MR T2 signal	Bt 12h	$\eta^2 = 0.1857$ $g = 0.933256$	X			
MRI	MR T2 signal	Bt 36h	$\eta^2 = 0.4704$ $g = 1.826791$	X			

MRI	MR T2 signal	DSS 5%	$\eta^2 = 0.03676$ $g = 0.386134$				X
FDG-PET	PUV _{Max norm}	Bt 3 h	$\eta^2 = 0.1124$ $g = 0.702109$				X
FDG-PET	PUV _{Max norm}	Bt 12h	$\eta^2 = 0.1893$ $g = 0.966266$	X			
FDG-PET	PUV _{Max norm}	Bt 24h	$\eta^2 = 0.2314$ $g = 1.093209$	X			
FDG-PET	PUV _{Max norm}	DSS 5%	$\eta^2 = 0.1607$ $g = 0.886772$	X			
FDG-PET	PUV _{Max norm}	Uracil 0.1M	$\eta^2 = 0.1609$ $g = 0.87674$	X			
FDG-PET	PUV _{Max norm}	Uracil 1.0M	$\eta^2 = 0.001127$ $g = 0.06767$				X
FDG-PET	PUV _{Max norm}	Microbacterium sp. (#3)	$\eta^2 = 0.1046$ $g = 0.697976$		X		
FDG-PET	PUV _{Max norm}	Enterococcus sp. (#4)	$\eta^2 = 0.004313$ $g = 0.146927$				X
FDG-PET	PUV _{Max norm}	E. coli	$\eta^2 = 0.07862$ $g = 0.592084$		X		X

* $\eta^2 = 0.01$ small effect, 0.06 moderate effect and 0.14 a large effect size

* $d/g = 0.2$ small effect, 0.5 moderate effect and 0.8 a large effect size

Table S15: Effect size of the used antibiotic treatments with the respective imaging modality; eta squared (η^2), Hedges' g or Cohen's d.

Imaging modality	Diagnostic finding	Treatment	Effect size of treatment η^2 , Hedges' g or Cohen's d (compared to Bt only)	Large effect*	Moderate effect*	Small effect*	No effect*
CT	CT gut wall thickness (post-iodixanol)	Bt + gentamicin 0.5 mg/ml	$\eta^2 = 0.08357$ $g = 0.661769$		X		
CT	CT gut wall thickness (post-iodixanol)	Bt + gentamicin 1 mg/ml	$\eta^2 = 0.3171$ $g = 1.59552$	X			
MRI	MR gut wall thickness (post-gadolinium)	Bt + gentamicin 0.5 mg/ml	$\eta^2 = 0.1850$ $g = 1.060247$	X			
MRI	MR gut wall thickness (post-gadolinium)	Bt + gentamicin 1 mg/ml	$\eta^2 = 0.3171$ $g = 1.616842$	X			

* $\eta^2 = 0.01$ small effect, 0.06 moderate effect and 0.14 a large effect size

* $d/g = 0.2$ small effect, 0.5 moderate effect and 0.8 a large effect size

Table S16: Effect size of the used anti-ROS treatments (DUOX) with the respective imaging modality; eta squared (η^2), Hedges' g or Cohen's d.

Imaging modality	Diagnostic finding	Treatment	Effect size of treatment η^2 , Hedges' g or Cohen's d (compared to uracil only)	Large effect*	Moderate effect*	Small effect*	No effect*
CT	CT gut wall thickness (post-iodixanol)	Uracil + DPI	$\eta^2 = 0.5555$ $d = 2.182477$	X			
CT	CT gut wall thickness (post-iodixanol)	Uracil + NAC	$\eta^2 = 0.5486$ $g = 2.153977$	X			
CT	CT signal density (post-iodixanol)	Uracil + DPI	$\eta^2 = 0.4111$ $g = 1.633457$	X			
CT	CT signal density (post-iodixanol)	Uracil + NAC	$\eta^2 = 0.3929$ $g = 1.573131$	X			
MRI	MR gut wall thickness (post-gadolinium)	Uracil + DPI	$\eta^2 = 0.1449$ $g = 0.813261$	X			
MRI	MR gut wall thickness (post-gadolinium)	Uracil + NAC	$\eta^2 = 0.3904$ $g = 1.558285$	X			
MRI	MR normalized T1 signal	Uracil + DPI	$\eta^2 = 0.1380$ $g = 0.801297$	X		X	
MRI	MR normalized T1 signal	Uracil + NAC	$\eta^2 = 0.1397$		X		

FDG-PET	PUV _{Max norm}	Uracil + DPI	$g = 0.783295$ $\eta^2 = 0.2312$	X	X
FDG-PET	PUV _{Max norm}	Uracil + NAC	$g = 1.088819$ $\eta^2 = 0.1299$	X	X
			$g = 0.754842$		X

* $\eta^2 = 0.01$ small effect, 0.06 moderate effect and 0.14 a large effect size

* $d/g = 0.2$ small effect, 0.5 moderate effect and 0.8 a large effect size

Table S17: Effect size of the used dexamethasone rescue after uracil treatment; eta squared (η^2), Hedges' g or Cohen's d.

Imaging modality	Diagnostic finding	Treatment	Effect size of treatment η^2 , Hedges' g or Cohen's d (compared to uracil +NaCl)	Large effect*	Moderate effect*	Small effect*	No effect*
CT	CT gut wall thickness (post-iodixanol)	Uracil + DEX	$\eta^2 = 0.6957$ $g = 2.88159$	X			
				X			

* $\eta^2 = 0.01$ small effect, 0.06 moderate effect and 0.14 a large effect size

* $d/g = 0.2$ small effect, 0.5 moderate effect and 0.8 a large effect size

Fig. S9: Intraobserver variability and retesting in the used imaging modalities. (A) Correlation of independent manual CT gut wall thickness measurements. (B) Bland Altman plot comparing the manual CT gut wall thickness measurements. (C) Correlation of two CT Signal density measurements. (D) Bland Altman plot comparing the Signal density measurements. (E) Correlation of two PUVMaxn measurements. (F) Bland Altman plot comparing the PUVMaxn measurements. Every data point represents a single animal.

- Is the methodology sound? Does the work meet the expected standards in your field?

See above. Experiments are indeed interesting and informative and interdisciplinary but the hypothesis to be tested and null hypothesis to be rejected do not drive to the “pitch”.

This can be corrected without additional experiments but must be corrected prior to publication.

Please see above our answer to your previous comment.

- Is there enough detail provided in the methods for the work to be reproduced?

Yes

Thank you.

REVIEWERS' COMMENTS

Reviewer #1 (Remarks to the Author):

The authors have responded to all my comments adequately, including new experiments. In particular, the additional experiments exhibiting the possibility of drug screening using Dex and the possibility of elucidating the function of AMPs are well-deserved results in showing the range of applications of this platform. I actually had felt some serious concerns about the versatility of this experimental platform, but now I think that this revision reinforces this point and dispels my concerns.

Reviewer #3 (Remarks to the Author):

The authors have done a thorough job responding to my concerns and addressed all issues with additional data. I recommend this paper for publication.

Reviewer #4 (Remarks to the Author):

All of my concerns were addressed. Please publish